

**Tracer-based source apportioning of atmospheric organic carbon and**
**the influence of anthropogenic emissions on secondary organic aerosol**
**formation in Hong Kong**
Yubo Cheng[1], Yiqiu Ma[1,2], Di Hu[1,2]
[1]State Key Laboratory of Environmental and Biological Analysis, Department of Chemistry, Hong Kong Baptist University,
Kowloon Tong, Kowloon, Hong Kong, P. R. China
[2]HKBU Institute of Research and Continuing Education, Shenzhen Virtual University Park, Shenzhen, 518057, P. R. China
*Correspondence to*: Di Hu (dihu@hkbu.edu.hk)



**Abstract.** Here we conducted comprehensive chemical characterization and source apportionment of 49 PM$_{2.5}$ samples
collected in Hong Kong. Besides the major aerosol constituents, 39 polar organic species, including 14 secondary organic
aerosol (SOA) tracers of isoprene, monoterpenes, β-caryophyllene, and naphthalene, were quantified using gas
chromatography-mass spectrometry (GC-MS). Six factors, i.e., SOA, secondary sulfate (SS), biomass burning (BB)/SOA, sea
salt, marine vessels, and vehicle emissions, were apportioned by positive matrix factorization (PMF) as the major sources of
ambient organic carbon (OC) in Hong Kong. The secondary formation, including OC from SOA, SS, and aging of BB plume,
was the leading contributor to OC (51.4%, 2.15±1.37 µgC m$^{-3}$) throughout the year. We then applied a tracer-based method
(TBM) to estimate the SOA formation from the photo-oxidation of four selected precursors, and monoterpenes SOA was the
most abundant. A Kintecus kinetic model was used to examine the formation channels of isoprene SOA, and the aerosol-phase
ring-opening reaction of isoprene epoxydiols (IEPOX) was found to be the dominant formation pathway. Consistently, IEPOX
tracers contributed 94% of total GC-MS quantified isoprene SOA tracers. The TBM-estimated SOC (SOC$_{TBM}$) and PMF-
resolved SOC (SOC$_{PMF}$) showed similar temporal trends; however, SOC$_{TBM}$ only accounted for 26.5% of SOC$_{PMF}$, indicating
a large fraction of ambient SOA was from other reaction pathways/precursors. Results of Pearson's R and multivariate linear
regression analysis showed that NOx processing played a key role in both daytime and nighttime SOA production in the region.
SOA formation through nighttime NO$_3$ oxidation of biogenic VOCs, especially monoterpenes, may have made a considerable
contribution to the SOA loading in Hong Kong. Moreover, sulfate had a significant positive linear relationship with SOC$_{PMF}$
and SS-related SOC, and particle acidity was significantly correlated with SOC from the aging of BB.
**1 Introduction**

Organic aerosol (OA) is a significant component of ambient fine particulate matter (PM$_{2.5}$). It accounts for 20%-60% of the

total PM$_{2.5}$ mass on a global scale (Kanakidou et al., 2005; Van Dingenen et al., 2004; Zhang et al., 2007), and even up to 90%
in rural areas (Kanakidou et al., 2005; Roberts et al., 2001; Zhang et al., 2007). OA is either directly emitted into the atmosphere
from natural (e.g., vegetative detritus, volcano activity) and anthropogenic sources (e.g., biomass burning (BB), vehicle
exhaust, cooking), or secondarily formed through the oxidation of biogenic and anthropogenic gas-phase precursors and the
subsequent partition process or particle-phase reactions (Gelencsér et al., 2007; Hildemann et al., 1996; Hu et al., 2010; Zheng
et al., 2014). Given the varying emission sources, meteorological conditions, and anthropogenic activities worldwide and their
influences on ambient OA composition, aerosol scientists have put many efforts to investigate the atmospheric processes of
OA and their primary and secondary sources, which aid the development of more targeted control policy of PM$_{2.5}$ pollution
(Hu et al., 2010; Huang et al., 2014; Schauer et al., 2007; Simoneit, 1999; Stone et al., 2009; Zheng et al., 2005). Huang et al.
(2014) applied positive matrix factorization (PMF) to apportion the sources of OA at four urban locations in China, i.e., Beijing,
Shanghai, Guangzhou, and Xi'an. They found that secondary formation accounted for a predominant fraction of OA (44-71%)
at all four sites. Hong Kong, a megacity located on the southern coast of China in the PRD region and a hub port for the South
Asian Pacific region, has its unique OC source characteristics. Hu et al. (2010) incorporated biogenic and anthropogenic SOA
tracers and some POA markers into PMF and resolved seven OA sources in Hong Kong. They found that 45% of OC in Hong
Kong during summertime was from secondary formation, and the number could reach up to 65% on sampling days under
regional pollution from the PRD area.
All these studies have illustrated the importance of secondary formation to OA in the ambient atmosphere. However, due
to SOA's complex chemical composition and formation mechanisms, a precise prediction of SOA load from individual
precursors at both regional and global scale is still challenging. An SOA tracer based method (TBM) has been developed to
partially solve this problem, which estimates the amount of SOA and SOC formed from the atmospheric oxidation of selected
VOCs (i.e., isoprene, monoterpenes, β-caryophyllene, toluene, and naphthalene) using the mass ratios of tracer-to-SOA/SOC
obtained from laboratory smog chamber experiments (Kleindienst et al., 2007, 2012). However, TBM can only capture SOC
formation from the above-listed VOC precursors, and it may underestimate the actual SOC levels in the ambient atmosphere
due to the lack of SOA tracer-to-SOC ratio values of a broader range of OA precursors. Therefore, besides the SOA tracer
based method, we have also applied PMF to evaluate the contributions of SOC and primary emissions to OA in the region.
Many studies have reported the observational evidence of biogenic SOA enhancement induced by anthropogenic emissions,
such as nitrogen oxides ($NO_X$) and sulfur dioxide ($SO_2$) (Huang et al., 2014; Xu et al., 2015; Rattanavaraha et al., 2016). $NO_x$
is one of the critical drivers of SOA formation through the photochemical oxidation of VOCs via peroxy radical pathways
(Finlayson-Pitts and Pitts, 2000). Nitrogen dioxide ($NO_2$) reacts with ozone ($O_3$) to form $NO_3$ radical, a critical nighttime gas
oxidant. Several laboratory studies have reported high SOA yields from the oxidation of biogenic VOCs (BVOCs) by $NO_3$
radical (Fry et al., 2009; Ng et al., 2008). Some field studies also revealed that SOA formation from $NO_3$ oxidation of BVOCs
occurs during both daytime and nighttime (Brown et al., 2013; Rollins et al., 2013). The effect of $SO_2$ on SOA formation was
often explained in the context of particle acidity in laboratory studies, which promotes SOA production through acid-catalyzed
heterogeneous reactions (Jang et al., 2002; Surratt et al., 2010). Sulfate was also suggested to enhance isoprene-SOA formation
by acting as the nucleophiles, providing active aerosol surface area, and through the salting-in effect (Rattanavaraha et al.,
2016; Xu et al., 2015). Recently, Wang et al. (2016) proposed a new sulfate formation pathway in aqueous aerosols through
$NO_2$ oxidation and ammonium neutralization, and synchronous enhancements of both nitrate and SOA production in aqueous
aerosols were reported. These laboratory and field monitoring studies have shown that the abundance and chemical nature of
ambient OA are significantly influenced by the complex interactions among source emissions, anthropogenic activities,
atmospheric physical/chemical processes, and meteorological conditions (An et al., 2019).
In this study, we collected 49 $PM_{2.5}$ samples at an urban site in Hong Kong during a whole year period. Concentration
levels of 39 polar organic species were quantified using gas chromatography-mass spectrometry (GC-MS), and their



temporal/meteorological variations were evaluated. With the input of SOA tracers and primary source markers into PMF, we
quantitatively assessed the contributions of various primary and secondary sources to OC. SOC formed from individual
biogenic (i.e., isoprenes, monoterpenes, and β-caryophyllene) and anthropogenic VOCs (i.e., naphthalene) were estimated
using the TBM. Finally, the impacts of anthropogenic pollutants (e.g., $NO_2$, $O_3$, $NO_3$, $SO_2$, and tropospheric odd oxygen ($O_X$))
and $PM_{2.5}$ constituents (e.g., sulfate, acidity, and liquid water content) on total and individual SOCs estimated by both TBM
and PMF were evaluated using Pearson's R analysis and multi-linear regression model. This study provides comprehensive
information on the sources of OA and SOA in Hong Kong as well as direct evidence of anthropogenic influences on the SOA
formation in the region, which may serve as the scientific basis for the formulation of the $PM_{2.5}$ mitigation policy in the region.
**2 Method**
**2.1. Sample collection**

The $PM_{2.5}$ samples were collected on the 12[th] floor of Science Tower in the Campus of Hong Kong Baptist University

(114°15E, 22°13N, ~40 m above the ground), which is a typical urban site. $PM_{2.5}$ samples were collected from September 6,
2011, to August 16, 2012, and a total of 49 samples were collected. A high-volume air sampler was used to collect $PM_{2.5}$ onto
a quartz fiber filter (20 cm × 25 cm) at a flow rate of 1.13 $m^3$ $min^{-1}$ for 24 h. The quartz fiber filters were prebaked at 550°C
for 24 h to remove organic contaminants. After sampling, the filters were immediately transferred to the laboratory and stored
at -18°C until analysis.
**2.2. Chemical analysis**

For EC and OC analysis, a 1 × 1 $cm^2$ filter was cut and analyzed using a thermal and optical transmittance aerosol carbon

analyzer (Sunset Laboratory, Tigard, OR, USA). Major ions (i.e., $Cl^-$, $NO_3^-$, $SO_4^{2-}$, $C_2O_4^{2-}$, $Na^+$, $Ca^{2+}$, $Mg^{2+}$, $K^+$, $NH_4^+$) were
identified and quantified by ion-chromatography (IC, DX500, Dionex, Sunnyvale, CA, USA). Vanadium (V) and Nickle (Ni)
were analyzed using an Agilent 7900 ICP-MS. Detailed analytical methods for the measurements of EC, OC, and ions were
described in our previous work (Hu and Yu, 2013; Ma et al., 2019).

Thirty-nine polar organic species were identified and quantified using an Agilent 7890A-5975C GC-MS with prior BSTFA

derivatization (N, O-Bis-(trimethylsilyl)trifluoroacetamide, with 1% trimethylchlorosilane, TMCS). For each aerosol sample,
20 $cm^2$ of the filter was cut into small pieces and sonicated for 10 min with 10 mL of distilled acetonitrile (HPLC grade); the
extraction was repeated three times. The extracts were combined and filtered through a Millipore 0.45-μm PTEE hydrophobic
Teflon filter into a 50 mL round flask, concentrated to ~ 0.5 mL by rotary evaporation, and transferred into a 5 mL reaction
vial. The round flask was rinsed with 1 mL of acetonitrile for three times, and the rinsing solvent was transferred into the
reaction vial as well. The final extract was blown to dryness under a gentle stream of pure nitrogen gas at 40 °C and then



derivatized with 100 μL of BSTFA and 50 μL of pyridine at 70 ℃ for 2 h. After the reaction vial cooled down to room
temperature, 30 μL of tetracosane-$d_{50}$ (internal standard, 50 μg mL$^{-1}$ in hexane) was added. The derivatives were analyzed by
GC-MS. Two microliters of the derivatized sample or standard were injected and separated on an HP-5MS capillary column
(30.0m×250μm×0.25μm, Agilent J&W). The temperature program and instrument settings were adapted from the method used
by Hu et al. (2008).

Saccharides, di- and tricarboxylic acids, 4-nitrocatechol, and cholesterol were identified and quantified using authentic

standards. The SOA tracers were identified using surrogate compounds with similar structures and functional groups (Hu et
al., 2008; Hu and Yu, 2013), and the detailed information was provided in Table 1. Recovery tests of these organic species
were carried out by spiking the mixture of standards onto blank quartz filters, followed by the same sample extraction and
analysis processes. Recoveries of the polar compounds were within the range of 80% to 120%. Analysis of hopanes has been
reported in our previous study (Ma et al., 2019). Four hopanes, including 17α,21β-hopane, 17α,21β-22R-homhopane,
17α,21β-22S-homhopane, and 17α,21β-30-norhopane, were measured using an Agilent 6890N-5975 GC-MS with thermal
desorption (TD) method. Recoveries of four hopane standards ranged from 83% to 98%.
**3 Results**

Hourly meteorological and air quality data (i.e., temperature, relative humidity (RH), $O_3$, $SO_2$, and $NO_2$) in the vicinity of

the    sampling    site    were    collected    by    Hong    Kong    Environmental    Protection    Department
(HKEPD) (http://envf.ust.hk/dataview/gts/current/). During the sampling period, the ambient temperature ranged from 14.52
to 31.01 ℃, with an annual average of 24.17±5.00 ℃. The daily average of RH ranged from 52.94% to 97.02%, with a yearly
average of 79.87±10.54%. Heavy rains are common in Hong Kong during summer, which effectively washes out the PM
pollutants.

Hong Kong is located at the south-east edge of the Pearl River Delta (PRD) region. PRD is a rapidly developing area with

intensive industrial activities. Air pollutants origin from the northern PRD region can travel together with air masses and
transport into Hong Kong. Same as in our previous study (Hu et al., 2010; Ma et al., 2019), we carefully examined the air mass
backward trajectories, the spatial distribution patterns of $SO_2$, the concentration levels of both $PM_{2.5}$ and $O_3$, and the synoptic
weather conditions during the sampling period. We then categorized all sampling days into three groups, i.e., days mainly
influenced by the regional pollution from the PRD region (regional days), days influenced by long-regional transport of air
mass from the northern and eastern China (LRT days), and days dominated by the locally generated pollutants (local days).

The gas pollutants, i.e., $O_3$, $NO_2$, $O_X$, and $SO_2$, showed significantly higher average concentrations on regional days than

those on LRT local days (Table 2). The annual mean concentrations of $O_3$, $NO_2$, and $SO_2$ were 14.85±8.69 ppb, 37.15±9.76
ppb, and 4.45±2.57 μg m$^{-3}$, respectively. Given that $O_3$ and $NO_2$ undergo a rapid photochemical conversion in the ambient





atmosphere, the tropospheric odd oxygen $O_X$ (the sum of $O_3$ and $NO_2$) was calculated as an indicator of atmospheric oxidation
capacity. As shown in Table 1, $O_X$ ranged from 49.72 to 145.90 µg m$^{-3}$, with a mean value of 99.31±27.42 µg m$^{-3}$, indicating
a high oxidation capacity of the Hong Kong atmosphere. The annual average concentrations of OC and EC were 4.18±2.37
and 1.02±0.54 µgC m$^{-3}$, respectively. Ambient OC levels observed on regional days (6.15±2.51 µgC m$^{-3}$) were about two times
higher than those on LRT and local days; as for EC, it exhibited relative constant levels throughout the year (0.14-2.75 µgC
m$^{-3}$). This confirms that EC is mainly emitted locally in Hong Kong, and OC has some regional sources. Moreover, the OC/EC
ratios of the collected samples ranged from 1.51 to 10.91, with an annual average value of 4.61, indicating secondary formation
could be a dominant source of OA in this region (Mancilla et al., 2015). Our previous study has observed that SOC contributed
45% of OC in Hong Kong during the summer of 2006 (Hu et al., 2008). Here the analysis was expanded to samples taken
during the 1-yr period to obtain a more comprehensive understanding of sources and their contributions to ambient OA in
Hong Kong, and the factors that impact ambient SOA formation.
**3.1 Characterization of SOA tracers and other polar oxygenated organic compounds**

The concentration levels of 39 organic species, including 14 SOA tracers, 12 saccharides, 11 di- and tricarboxylic acids, 4-

nitrocatechol, and cholesterol, under different meteorological conditions, were listed in Table 1.
**3.1.1 SOA tracers of isoprene, monoterpenes, β-caryophyllene, and naphthalene**

Seven isoprene SOA (Isop_SOA) tracers, i.e. 2-methylglyceric acid, two methyltetrol isomers (2-methylthreitol and 2-

methylerythritol), three $C_5$-alkene triol isomers (cis-2-methyl-1,3,4-trihydroxy-1-butene, 3-methyl-2,3,4-trihydroxy-1-butene,
and tans-2-methyl-1,3,4-trihydroxy-1-butene), and 3-meTHF-3,4-diols (including both cis- and trans-3-
methyltetrahydrofuran-3,4-diols) were identified and quantified. The sum of all Isop_SOA tracers ranged from 1.67 to 117.17
ng m$^{-3}$, with the annual mean value of 22.78±26.06 ng m$^{-3}$. Among the Isop_SOA tracers, methyltetrols and $C_5$-alkene triols
were the most abundant, and they are suggested to be formed through the acid-catalyzed ring-opening reactions of IEPOX
under low-$NO_X$ condition (Chan et al., 2010; Surratt et al., 2010). Higher concentrations of Isop_SOA tracers were measured in
summer and autumn than in winter and spring. This could be caused by the higher temperature, stronger solar radiation, and
higher emission of isoprene in summer and autumn than in the other two seasons, which promoted the SOA formation from
isoprene. This seasonal pattern is consistent with what were observed in other studies (Ding et al., 2012; Kleindienst et al.,
2007; Lewandowski et al., 2008). However, if we compare the levels of isoprene tracers monitored at different sites during
summer, the total amount of Isop_SOA tracers measured in Hong Kong was about five times lower than those measured in
several cities in the U.S. and a rural site (WQS site) in the PRD area (Ding et al., 2012; Kleindienst et al., 2007; Lewandowski
et al., 2008). This may be due to the different levels of isoprene, OH radical, and $NO_X$ at these sampling sites. The 3-MeTHF-

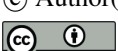



3,4-diols, tracers formed through the intermolecular rearrangement of isoprene epoxydiols (IEPOX) under acidic conditions,
was identified in Hong Kong PM$_{2.5}$ samples for the first time. It has an annual mean concentration of 0.23±0.10 ng m$^{-3}$, which
was about 70 times lower than that in Birmingham, U.S. (Rattanavaraha et al., 2016), but was comparable to what was observed
at the WQS site in the PRD area (He et al., 2018). 2-Methylglyceric acid, an isoprene tracer formed from methacrylic acid
epoxide (MAE) and hydroxymethyl-methyl-α-lactone (HMML) under high-NO$_x$ conditions (Lin et al., 2013; Nguyen et al.,
2015), presented a quite different temporal trend from those of the other six Isop$_{SOA}$ tracers, with the highest concentration in
winter, and then autumn, summer, and spring. Chamber studies suggested that MAE is an oxidation product resulting from the
OH addition to methacryloylperoxynitrate (MPAN) and its production is temperature dependant (Roberts and Bertman, 1992;
Worton et al., 2013). Under higher temperatures, the loss of MPAN is dominated by thermal decomposition, which does not
produce SOA tracers through the NO/NO$_2$ pathway. Under lower temperature, thermal decomposition of MPAN is limited and
more MPAN reacts with OH to generate MAE. Therefore, the lower temperatures in winter would favor the production of
MAE and the MAE Isop$_{SOA}$ tracers, such as 2-methylglyceric acid. Moreover, all Isop$_{SOA}$ tracers exhibited higher
concentrations on regional days than LRT and local days. On regional days, air masses transported from the PRD area worsened
the air quality in Hong Kong, and the higher levels of gaseous pollutants, e.g., O$_3$, NO$_2$, O$_X$, and SO$_2$ (Table 2), promoted SOA
formation.

Generally speaking, at an urban location with anthropogenic NO$_X$ emissions from automobiles and power plants, the

generation of Isop$_{SOA}$ tracers from the MAE NO/NO$_2$ pathway should be more favored than the HO$_2$ IEPOX channel.
However, in this study, 94% of the total mass of the quantified Isop$_{SOA}$ tracers were produced through the IEPOX HO$_2$
pathway. A similar phenomenon was observed at the WQS site in the PRD region (He et al., 2018). Therefore, to better
understand the influences of environmental factors on isoprene SOA formation in the region, we applied the kinetic models
described by Eddingsaas et al. (2010), Worton et al. (2013), and Birdsall et al. (2014) to investigate the fate of both IEPOX
and MAE in the atmosphere. Besides their degradation through acid-catalyzed ring-opening reactions on particles, IEPOX and
MAE can also be oxidized in the gas phase or removed by dry deposition (Eddingsaas et al., 2010). We applied the Kintecus
kinetic model to quantitatively evaluate the fractions of these two Isop_SOA intermediates that undergo gas-phase oxidation,
aerosol-phase acid-catalyzed ring-opening reaction, and dry deposition processes. Details of the model calculations were
provided in the appendices.

Figure 1 showed the comparison of the three elimination processes of IEPOX and MAE during the sampling period in

Hong Kong. Given the high volatility of MAE (vapor pressure: 9.2×10$^{-5}$ atm) (Worton et al., 2013), it has a low tendency to
partition onto the particle phase and its uptake onto aqueous particles is mainly governed by Henry's law constant (k$_H^{cp}$).
Worton et al. (2013) estimated the k$_H^{cp}$ value of MAE to be 7.5×10$^6$ M atm$^{-1}$, which is 20 times lower than that of IEPOX
(1.3×10$^8$ M atm$^{-1}$). Moreover, Riedel et al. (2015) suggested that the heterogeneous reactive uptake coefficient of MAE (γ =



$4.9 \times 10^{-4}$) through the ring-opening reaction was a factor of 30 lower than that of IEPOX. Therefore, as shown in Figure 1,
MAE was primarily eliminated by dry deposition (> 80%) in the gas phase, and only a trivial fraction was degraded through
the ring-opening reactions (≤ 2%). Our results on the fate of MAE were similar to those observed at the University of
California-Blodgett Forest Research Station (UC-BFRS) (Worton et al., 2013). However, our results on the relative
contributions of these three degradation pathways to IEPOX loss were quite different from theirs, indicating a more sensitive
response of IEPOX than MAE to the change of environmental oxidants and conditions. Given the high liquid water content
(LWC; mean: $57.20 \pm 37.15$ μg m$^{-3}$) and particle acidity ($H_p^+$; mean: $-0.28 \pm 0.42$) of PM$_{2.5}$ samples in this study (Table 2),
particle-phase ring-opening reaction ($F_{rop}$) was the dominant degradation pathway of IEPOX in the Hong Kong atmosphere
(average: 97.6%), and its loss through dry deposition and gas-phase oxidation is almost negligible. The $F_{rop}$ of IEPOX reported
by Worton et al. (2013) was only 0.02%, mainly due to the much lower LWC (mean: 0.4 μg m$^{-3}$) and weaker $H_p^+$ (pH mean:
4.4) of their PM$_{2.5}$ samples. These results demonstrated that particle-phase LWC and $H_p^+$ played a more significant role in the
atmospheric degradation of IEPOX than MAE. Results from the kinetic model simulation were strongly supported by the
experimental finding of IEPOX tracers as the dominant Isop$_{SOA}$ tracers measured in Hong Kong. The average ratio of IEPOX
tracers to MAE tracers was 16.54 (ranged from 3.00 to 71.58), and the average value of $F_{rop\text{-}IEPOX}/F_{rop\text{-}MAE}$ was 191.92,
confirming that the IEPOX HO$_2$ channel is the major formation pathway of isoprene SOA in the region.

Five SOA tracers of monoterpenes (Mono$_{SOA}$), i.e., 3-hydroxyglutaric acid, 3-hydroxy-4,4-dimethylglutaric acid, 3-

methyl-1,2,3-butanetricarboxylic acid, 3-isopropylpentanedioic acid, and 3-acetyl pentanedioic acid, were identified and
quantified. Their summed concentrations ranged from 2.54 to 32.57 ng m$^{-3}$, with an annual average value of $10.76 \pm 8.04$ ng m$^{-}$
$^3$, comparable to that reported at the WQS site in the PRD region but lower than that measured in the U.S. (Ding et al., 2012;
Kleindienst et al., 2007; Lewandowski et al., 2008). All Mono$_{SOA}$ tracers showed the highest level on regional days (mean:
$18.00 \pm 9.28$ ng m$^{-3}$), followed by LRT (mean: $10.31 \pm 7.33$ ng m$^{-3}$) and local days (mean: $6.41 \pm 2.75$ ng m$^{-3}$) (Table 1). Although
a higher emission and faster photochemical degradation of monoterpenes are expected in summer due to the intense solar
radiation and high temperature, higher levels of Mono$_{SOA}$ tracers were monitored in autumn and winter than in the other two
seasons, similar to what observed at the WQS site (Ding et al., 2014). This seasonal trend of monoterpene SOA tracers may
be partly due to the lower mixing height and temperature during autumn/winter, which favored the partition of Mono$_{SOA}$
tracers onto the aerosol phase. Moreover, most of the regional days were identified in autumn and winter. The higher levels of
NO$_X$, O$_3$, O$_X$, and SO$_2$ on regional days (Table 2) are also responsible for the enhanced monoterpene SOA production in autumn
and winter. Among Mono$_{SOA}$ tracers, 3-hydroxyglutaric acid (3HGA) was the most abundant, contributing ~60% of the total
mass of Mono$_{SOA}$ tracers. Smog chamber experiments showed that the production yield of 3-methyl-1,2,3-butanetricarboxylic
acid (MBTCA) from α-pinene/NOx oxidation was significantly higher than those from the β-pinene/NOx and d-limonene/NO$_X$
experiments (Jaoui et al., 2005). Therefore, the ratio of 3HGA/MBTCA was used as a criterion to differentiate SOA from α-



pinene and other monoterpenes (Ding et al., 2014). The value of this ratio was obviously higher on regional days (8.58±2.69)
than those on LRT (6.64±3.63) and local days (5.62±3.14), indicating that monoterpenes other than α-pinene, such as β-pinene
and d-limonene, might have a more significant contribution to SOA on regional days in the region.

Beta-caryophyllinic acid is the SOA (Cary$_{SOA}$) tracer of β-caryophyllene, and it ranged from 0.49 to 5.82 ng m$^{-3}$, with

an average annual mean value of 1.53±1.07 ng m$^{-3}$. Similar to the other SOA tracers, β-caryophyllinic acid showed the highest
concentrations on regional days (mean: 2.33±1.21 ng m$^{-3}$) than LRT (1.73±1.16 ng m$^{-3}$) and local days (0.94±0.41 ng m$^{-3}$)
(Table 1). For its seasonal trend, β-caryophyllinic acid also exhibited the highest concentration in autumn and winter than the
other two seasons. The SOA tracer of toluene, 2,3-dihydroxy-4-oxopentanoid acid, was undetectable in this study, mainly due
to its trace level in the Hong Kong atmosphere (Hu et al., 2008) and the limited sensitivity of GC-quadruple MS. Even in our
previous study on a batch of summer PM$_{2.5}$ samples using a more sensitive GC-ion trap MS, it was barely quantified with a
concentration of less than 1 ng m$^{-3}$ in most samples (Hu et al., 2008). Phthalic acid was suggested as the SOA tracer of
naphthalene, given its abundance in both naphthalene-SOA and ambient OA (Kleindienst et al., 2012). With the awareness of
the potential uncertainties, e.g., the primary origin of phthalic acid from biomass burning, we adopted phthalic acid as the SOA
tracer of naphthalene representing the SOA formation from anthropogenic VOCs. The concentration levels of phthalic acid
ranged from 0.80 to 16.42 ng m$^{-3}$, with an average of 4.31±3.39 ng m$^{-3}$. Similar to the other SOA tracers, it also showed the
highest concentrations on regional days (7.16±3.61 ng m$^{-3}$) than LRT (4.97±3.30 ng m$^{-3}$) and local days (2.26±1.38 ng m$^{-3}$)
(Table 1).

### 237   3.1.2 Saccharides and dicarboxylic acids

Twelve saccharides, i.e., levoglucosan, arabitol, fructose, meso-erythritol, sucrose, galactosan, mannitol, sorbitol,

galactose, glucose, xylose, and xylitol, have been quantified. Of the 12 saccharides, levoglucosan, the tracer of BB, was by far
the most abundant (range: 0.64-474.15 ng m$^{-3}$; mean: 75.02±111.43 ng m$^{-3}$). It showed the highest levels on regional days
(about 6 times higher than that on local days), especially during winter when BB activities in the PRD region were most
frequent. Two primary saccharides, i.e., fructose and xylose, also exhibited the highest levels on regional days. They showed
good correlations with levoglucosan (R$^2$= 0.65 and 0.93), suggesting that they could be from BB as well.

Among the identified dicarboxylic acids, oxalic acid was the most abundant, followed by terephthalic acid, phthalic acid,

malic acid, succinic acid, and others. Most dicarboxylic acids, including the five most abundant ones, showed higher levels on
regional days; they were found with higher levels in winter and autumn as well. This temporal trend is similar to what we have
observed for Mono$_{SOA}$ tracers and most saccharides, indicating that regional pollution had a dominant influence on the
abundance of both primary and secondary aerosols in Hong Kong, far exceeding the influence of other environmental
parameters, such as temperature and solar radiation. Atmospheric dicarboxylic acids have various sources. For example, oxalic



acid was suggested to be secondarily formed from biogenic emissions and anthropogenic sources (e.g., BB and automobile
exhaust) through both gas-phase reactions and in-cloud processing (Yu et al., 2005). Malic acid was suggested to be the photo-
degradation product of both succinic acid and biogenic SOA compounds (Hu and Yu, 2013). In this study, malic acid was
found to be strongly correlated with 3HGA ($R^2$=0.96) and $\Sigma$Mono_$_{SOA}$ tracers ($R^2$=0.95) throughout the year, providing more
evidence to the hypothesis that malic acid is a late-stage oxidation product of BVOCs, especially monoterpenes (Hu and Yu,
2013). Ambient terephthalic acid was mainly directly emitted from plastic wastes incineration (Simoneit et al., 2005) and was
used as a marker of waste incineration.

Besides dicarboxylic acids, two benzenetricarboxylic acids (i.e., 1,2,3- and 1,2,4-benzenetricarboxylic acids), 4-

nitrocatechol, and cholesterol were also quantified. The two benzenetricarboxylic acids were suggested to be the photo-
degradation products of polycyclic aromatic hydrocarbons (PAHs) emitted from the combustion activities (Kautzman et al.,
2010). We have previously identified them in the water-soluble humic-like substances (HULIS) extracts of $PM_{2.5}$ samples
collected in Beijing and Hong Kong (Ma et al., 2018, 2019). The annual mean concentrations of 1,2,3- and 1,2,4-
benzenetricarboxylic acids measured in this study were 2.27±1.97 ng m$^{-3}$ (range: 0.47-9.50 ng m$^{-3}$) and 3.13±2.68 ng m$^{-3}$ (0.47-
12.54 ng m$^{-3}$), respectively, which were comparable to what measured at the other four sites in the PRD region (He et al., 2018).
The 4-nitrocatechol, which was secondarily generated from the photo-oxidation of naphthalene, was suggested as the tracer of
atmospheric aging of BB plume (Kitanovski et al., 2012). It strongly correlated with levoglucosan ($R^2$=0.88) and exhibited
higher levels on regional days and during winter, which further confirmed its BB origin in the region. Therefore, the two
benzenetricarboxylic acids and 4-nitrocatechol were included in PMF analysis as the SOA tracers of BB aging.
**3.2 Source apportionment of organic aerosols**

In this study, PMF analysis was performed to determine the major OA sources and quantify their contributions to OC.

Eighteen species were input into PMF, including EC, OC, Ni, V, major ions, and various primary and secondary organic tracers.
Given their similar origins, some organic tracers were lumped together, and the lumped species were used as the fitting species
in PMF. They were (1) $C_5$-alkene triols, sum of the three $C_5$-alkene triols isomers; (2) IsopT, the sum of two methyltetrol
isomers and 2-methyl glyceric acid; (3) MonoT, the sum of the five monoterpenes SOA tracers; and (4) Hopane, the sum of
the four hopanes. Since $C_5$-alkene triols were not in the SOA tracers list of the TBM (Kleindienst et al., 2007), the lumped $C_5$-
alkene triols were used as a separated fitting species in PMF. PMF solutions were tested with 4 to 8 factors. A hundred base
runs were performed in each modeling run, and the run with the minimum Q value was selected. The uncertainty values of
each input species were calculated using the method described in our previous studies (Hu et al., 2010; Ma et al., 2016), which
were set to be 20% of the mean concentrations for OC and EC, and 40% of mean values for cations, anions, and all organic
species. An extra modeling uncertainty of 10% was used to account for possible temporal changes in the source profiles. The



$Q_{Robust}/Q_{True}$ ratio was 1.00, and scaled residuals were normally distributed between -0.2 and 0.2, indicating no influence of
outliers on the solution. A hundred bootstrap runs were performed with a minimum correlation R-value to examine the base
run solution's stability and uncertainty. All bootstrapped factors were explicitly mapped to factors resolved in base solution
with no exception. In the displacement (DISP) assessment, no error was found, and the drop of Q value was less than 1%,
suggesting a stable solution. No swap factor appeared at $dQ_{max}=4$, indicating there was no considerable rotational ambiguity
in the solution. Rotations were introduced to the solutions by adjusting the FPEAK value from -1 to +1, and the non-rotated
solutions (FPEAK=0.0) were considered to be the most interpretable ones. Moreover, a strong linear correlation between the
measured and PMF-predicted OC ($OC_{PMF}$) ($R^2=0.92$) was observed, which also suggested a reliable PMF solution. Moreover,
a strong linear correlation between the measured and PMF-predicted OC ($OC_{PMF}$) ($R^2=0.92$) was observed, which also
suggested a reliable PMF solution.

As shown in Figure 2, the first factor was distinguished by high loadings of oxalate and biogenic SOA tracers, suggesting

the secondary origin of this source. The second factor was dominated by large amounts of $SO_4^{2-}$ and $NH_4^+$, suggesting the
process of secondary sulfate formation. In the third factor, about 90% of levoglucosan was resolved into it, accompanied by
4-nitrocatechol, phthalic acid, and the two benzenetricarboxylic acids, indicating both the primary emission and aging of BB
plume. Therefore, this factor was defined as BB and SOA (BB/SOA). The fourth factor was identified as vehicular emissions
due to the large amounts of hopanes and EC resolved. The fifth factor has large amounts of Ni and V, which are signatures of
residual oil combustion from the marine vessel (Viana et al., 2009). It is well known that Hong Kong is one of the busiest
container ports globally, which handles 50% of the PRD's total cargo throughput. Therefore, the fifth factor was identified as
marine vessels. The sixth factor has a high loading of $Na^+$, $Mg^{2+}$, and $Ca^{2+}$, indicating the sea salt source.

The two leading sources contributing to ambient OC in Hong Kong were BB (including both primary emission and aging

process, $OC_{BB}$: 27.9%, 1.17±1.99 µgC m$^{-3}$) and SOA ($SOC_{SOA}$: 27.5%, 1.15±0.82 µgC m$^{-3}$), followed by marine vessels
($OC_{marine}$: 15.6%, 0.65±0.58 µgC m$^{-3}$), SS ($SOC_{SS}$: 14.5%, 0.60±0.46 µgC m$^{-3}$), vehicle emissions ($OC_{vehicle}$: 10.5%, 0.44±0.42
µgC m$^{-3}$), and sea salt ($OC_{sea}$: 4.0%, 0.17±0.19 µgC m$^{-3}$) (Table 2 and Fig. 3). Since a fraction of SOA from the aging of BB
($SOC_{BB}$) was resolved into the BB/SOA factor , we calculated $SOC_{BB}$ using the following equation:
$$SOC_{BB} \;\; = OC_{BB} - \frac{[LEVO_{BB}]}{0.082} \qquad\qquad (1)$$

where $OC_{\_BB}$ and $[LEVO_{BB}]$ are the amounts of OC and levoglucosan resolved in the BB/SOA factor. Using levoglucosan

as the tracer of primarily emitted BB OA, we calculated the amounts of POC from BB ($POC_{BB}$) by dividing $[LEVO]_{BB}$ with
0.082, where 0.082 is the average ratio of levoglucosan to POC from the burning of major types of Chinese cereal straws (i.e.,
rice, wheat, and corn) obtained in the combustion chamber experiments (Zhang et al., 2007). As cereal straws are one of the
most common BB fuels in China, the above ratio (0.082) has been used to estimate BB contribution to POC in both Beijing
(Zhang et al., 2008) and Hong Kong (Sang et al., 2011). Therefore, it was adopted to calculate $POC_{BB}$ in this study.



Based on PMF results, the source-specific contributions to OC were presented in Table 2 and demonstrated in Fig. 3. The
total SOC apportioned by PMF ($SOC_{PMF}$), i.e., the sum of $SOC_{SOA}$, $SOC_{SS}$, and $SOC_{BB}$, accounted for 51.4% (2.15±1.37 μgC
m$^{-3}$) of OC in Hong Kong, with the secondary organic-rich sources (i.e., $SOC_{SOA}+SOC_{BB}$) contributing 36.9% (1.54±1.13 μgC
m$^{-3}$) of total OC. Huang et al. (2014) also reported that secondary organic-rich sources accounted for 30-40% of OC in
Guangzhou, another PRD site. A higher level of $SOC_{PMF}$ and its contribution to OC were observed on regional days (3.27±1.18
μgC m$^{-3}$, 57.4%) than on LRT (2.36±1.54 μgC m$^{-3}$, 53.0%) and local days (1.36±0.81 μgC m$^{-3}$, 43.6%). An even starker
difference in the amounts of $SOC_{BB}$ between regional and local days was observed, which was eight times higher on the
regional days. This suggested that non-local sources were the dominant contributors to $SOC_{BB}$. BB activities were intensive in
the PRD region, especially during fall and winter. On regional days, freshly emitted and aged gaseous and aerosol phase
pollutants from the open burning of rice straws and other crops were transported from the northern PRD region into Hong
Kong (Hu et al., 2010). Huang et al. (2014) examined the aging of BB plume at low temperatures. They found that the
production of BB SOA was rapid at a typical OH radical concentration of wintertime China, and the amount of BB SOA may
exceed BB POA in 4-14 h even at -10 °C. Given that the average temperature in Hong Kong during autumn and winter was
26.15 °C and 17.76 °C, the formation of BB SOA should be even fastly achieved during the regional transport. As expected,
$SOC_{SOA}$ also showed a higher average concentration on regional days (1.75±0.75 μgC m$^{-3}$) than on LRT (1.14±0.82 μgC m$^{-3}$)
and local days (0.78±0.65 μgC m$^{-3}$), which is consistent with the trends of all SOA tracers. Although SOC from secondary
inorganic-rich source ($SOC_{SS}$) exhibited the highest levels (0.82±0.38 μgC m$^{-3}$) on regional days as well, its contribution to
OC was relatively stable under the three synoptic conditions (Fig. 3). Several studies showed that $SO_2$ transported from the
northern PRD region promoted secondary sulfate formation in Hong Kong through both gas-phase and in-cloud oxidation
pathways (Lu and Fung, 2016; Yu et al., 2005; Yuan et al., 2006). A recent study proposed that the sulfate formation in aqueous
aerosols through $NO_2$ oxidation and ammonium neutralization can simultaneously enhance the production of both nitrate and
SOA (Wang et al., 2016), which helps explain the considerable amount of $SOC_{SS}$ apportioned.
OC from the four primary sources, i.e., $POC_{BB}$, $OC_{marine}$, $OC_{vehicle}$, and $OC_{sea}$, accounted for 48.6% of total OC throughout
the year. Similar to $SOC_{BB}$, $POC_{BB}$ showed a higher level (1.38±1.75 μgC m$^{-3}$) on regional days due to a large number of
emissions from BB activities in the northern PRD area. $OC_{vehicle}$ remained a higher contribution on local days (15.6%,
0.49±0.46 μgC m$^{-3}$), consistent with our previous finding that vehicle emission is a local pollution source (Hu et al., 2010).
Similarly, marine vessels accounted for a greater amount and larger fraction of OC on local days (32.0%, 1.00±0.63 μgC m$^{-3}$)
than LRT (5.2%, 0.23±0.19 μgC m$^{-3}$) and regional days (6.5%, 0.37±0.21 μgC m$^{-3}$). On local days, the southeastern to
southwestern wind brought pollutants from residual oil combustion from the ocean into Hong Kong, leading to a higher
$OC_{marine}$.
In summary, both secondary aerosol sources and air mass origins play important roles in atmospheric OC in Hong Kong.



On regional days, air mass transported from the northern PRD area brought large amounts of air pollutants into Hong Kong,
which promoted the SOA production from both anthropogenic emissions and BVOCs and resulted in a fraction of 57.4% of
OC being secondarily formed. On the other hand, local sources, including vehicle emissions and marine vessels, became more
critical and significantly contributed to OC (56.4%) on local days.
**3.3 Estimation of SOC origin**

To better understand the SOA precursors and their contributions to SOA/SOC in the region, we adopted a tracer-based

method (Kleindienst et al., 2007, 2012; Offenberg et al., 2007) to estimate the SOA/SOC formation from a group of selected
biogenic and anthropogenic hydrocarbons, i.e., isoprene, monoterpenes, β-caryophyllene, and naphthalene. The mass ratio of
tracer compounds to the total SOC ($f_{SOC}$) generated from individual VOC precursors was derived from smog chamber
experiments (Kleindienst et al., 2007; Offenberg et al., 2007). By assuming the same $f_{SOC}$ value of the precursor under smog
chamber conditions and in ambient air, one can use the quantified SOA tracer concentrations to estimate the amount of SOC
from that precursor in the real atmosphere. It has been well noted that results obtained from this tracer-based method are subject
to potential uncertainties from various aspects, e.g., the larger variation of precursor concentrations and more complicated
environmental conditions in the real atmosphere than in smog chamber experiments, the decay of some tracer compounds
during transport, mismatch of ambient and smog chamber generated SOA compositions, using surrogates other than ketopinic
acid for the quantification of tracer compounds, and so on (Ding et al., 2014; Hu et al., 2008; Kleindienst et al., 2012, 2007).
However, using the tracer-based method, we can at least have a rough estimation of the key SOA precursors in the region,
their contributions to ambient OC, and the amount of SOC from unknown precursors. Wang et al. (2013) noted that the SOA
tracer-based method would significantly underestimate SOC_Mono in the PRD region. Ding et al. (2014) gave a reasonable
explanation that the mismatch of monoterpene tracers measured in ambient air and used to derive $f_{SOC}$ of monoterpenes in
chamber studies may increase the uncertainty of SOC_Mono. Thus they picked the five Mono_SOA tracers measured in their
samples and derived the $f_{SOC}$ and $f_{SOA}$ values using the SOA tracers data and SOA/SOC concentrations reported by Offenberg
et al. (2007). In this study, we only measured five out of nine monoterpene SOA tracers in Offenberg et al. 's (2007) study.
Similar to Ding et al. (2014), to lower the uncertainty induced from the mismatch of SOA tracer compositions, we derived a
$f_{SOC\_mono}$ value of 0.047 based on Offenberg et al.'s experimental data (2017) and applied it to estimate SOC_Mono. Many
research groups have adopted this tracer-based method to assess SOC productions from the five studied VOCs at various
locations in the world, and reasonable results have been obtained (Ding et al., 2012; Fu et al., 2014; Hu et al., 2008; D. Hu and
Yu, 2013; Kleindienst et al., 2012, 2007; Lewandowski et al., 2008).

As shown in Table 2, SOC estimated by the tracer-based method (SOC_TBM) ranged from 0.11 to 1.53 µgC m$^{-3}$ in Hong

Kong, accounting for 3.8% to 22.7% of ambient OC levels. It exhibited the same trend as OC and SOC_PMF, i.e., with higher



concentrations on regional days ($0.81\pm0.35$ µgC m$^{-3}$) than on LRT ($0.50\pm0.29$ µgC m$^{-3}$) and local days ($0.28\pm0.13$ µgC m$^{-3}$).
Similar to our previous study, monoterpenes were found to be the most significant SOC contributor in the region, with
SOC$_{\_Mono}$ ranging from 0.05 to 0.69 µgC m$^{-3}$ and having an average concentration of $0.23\pm0.17$ µgC m$^{-3}$. SOC$_{\_Iso}$ and SOC$_{\_Cary}$,
on the other hand, were about three times smaller than SOC$_{\_Mono}$ and were $0.08\pm0.09$ µgC m$^{-3}$ and $0.07\pm0.05$ µgC m$^{-3}$,
respectively. Smog chamber experiments have been carried out to study the SOA yields from ·OH oxidation, ozonolysis, and
nitrate radical (NO$_3$) oxidation of monoterpenes and isoprene, and monoterpenes were found to be more effective in SOA
production than isoprene (Lee et al., 2006a, 2006b). Highly oxygenated organic molecules with low and extremely low
volatility were formed from the oxidation of monoterpenes and observed in both laboratory experiments and field
measurements (Ehn et al., 2014; Jokinen et al., 2015; Zhang et al., 2018). Moreover, a synergistic O$_3$ + OH oxidation pathway
of monoterpenes was recently proposed, which leads to the formation of extremely low-volatility oligomers and may result in
even larger monoterpene SOA yields in the real atmosphere than what observed in the smog chamber experiments (Kenseth et
al., 2018). Tsui et al. ( 2009) reported a total BVOC emission of $8.6\times10^9$ gC yr$^{-1}$ in Hong Kong, with 40% from monoterpenes
and 30% from isoprene. The remaining 30% could be sesquiterpenes (e.g., β-caryophyllene) or other BVOCs. Therefore, the
predominance of monoterpenes SOA in BVOCs-derived SOC is likely due to the combined effects of their high SOA yields
and large emissions in the region. Like the SOA tracers, SOC from the four precursors all showed the highest level on regional
days than those on LRT and local days (Table 2). On regional days, large amounts of VOC precursors and gaseous oxidants
could be brought into Hong Kong through the regional transport of air masses from northern PRD and oxidized along the way.
Conversely, on local days, the ocean breeze brings clean air masses from the South China Sea into Hong Kong, leading to a
dilution effect of local air pollution. These results highlight that mass origins play an important role in the SOC formation from
both biogenic and anthropogenic VOCs. Given that SOC$_{TBM}$ is calculated based on the concentration levels of individual SOA
tracers measured in the ambient aerosols, it is reasonable that SOC attributed to each VOC precursor showed the same
meteorological variations as their SOA tracers.
We observed similar temporal trends between SOC$_{PMF}$ and SOC$_{TBM}$ (R$^2$=0.71). However, SOC$_{TBM}$ only accounted for
26.5% of SOC$_{PMF}$, suggesting SOC must have been underestimated by the tracer-based method. A reasonable explanation is
that secondary formation from nighttime reactions, multi-phase reactions, and other SOA precursors are not considered in the
SOA tracer-based method. Because parameters used in the tracer-based method were derived from pure gas-phase photo-
oxidation of VOC precursors in smog chambers (Kleindienst et al., 2007, 2009). Therefore, it is better to be used as a
complementary method with PMF in the source apportionment study of ambient OC, especially SOC.
**3.4 Effects of anthropogenic influences on secondary aerosol formation**
Increasing evidence from laboratory studies and ambient observations has shown that anthropogenic emissions can





significantly affect SOA formation from terpenoids through multiple chemical processes in both daytime and nighttime (Xu et
al., 2013; Zhang et al., 2018). We conducted the Pearson's R correlation analysis of all SOC terms (i.e., $SOC_{\_Iso}$, $SOC_{\_Mono}$,
$SOC_{\_Cary}$, $SOC_{\_Nap}$, $SOC_{TBM}$, $SOC_{PMF}$, $SOC_{BB}$, $SOC_{SOA,}$ and $SOC_{SS}$) with $O_3$, $NO_2$, $SO_2$, $O_X$, $NO_3$, sulfate, particle acidity ($H_P^+$),
and particle liquid water content ($LWC_P$) (Table 3). Details on the calculation of $H_P^+$ and $LWC_P$ were presented in Appendix
B. Since $NO_3$ was not directly monitored at HKEPD stations, its mixing ratio was estimated using the following equation:

$$[NO_3] = \frac{k_1[O_3][NO_2]}{\sum k_i[VOC_i]} \qquad\qquad\qquad (2)$$

The numerator is the production of $NO_3$ ($p[NO_3]$) from $O_3$ and $NO_2$, and the denominator is the reactivity of $NO_3$ for $NO_3$-
VOCs reactions. From the IUPAC database, we obtained the temperature-dependent expression of $k_1$ ($cm^3$ molecules$^{-1}$ s$^{-1}$, the
production rate constant of $NO_3$) as 1.4E-13EXP(-2470/T), where T is the ambient temperature in Kelvin. Therefore, using $k_1$
and the measured concentration levels of $[O_3]$ and $[NO_2]$, we calculated $p[NO_3]$ (Table 2). Brown et al. (2016) reported a $NO_3$-
VOCs reactivity of $6.5\pm6.8\times10^{-3}$ s$^{-1}$ in Hong Kong with a corresponding $NO_3$ lifetime of 2.5 min. $NO_3$ was then calculated as
the ratio of $p[NO_3]$ to this $NO_3$ reactivity value, and an annual mean level of 70±47 ppt was estimated.
As we mentioned earlier, $O_X$ is an indicator of atmospheric oxidation capacity. Five SOC terms, i.e., $SOC_{\_Mono}$, $SOC_{\_Nap}$,
$SOC_{TBM,}$ $SOC_{SOA}$, and $SOC_{PMF}$, showed significant positive correlations with $O_X$, especially $SOC_{SOA}$ and $SOC_{PMF}$ (R>0.7,
P<0.01). However, only $SOC_{SOA}$ and $SOC_{SS}$ were found to be significantly correlated with $O_3$ (R>0.50, P<0.01). As for $NO_2$,
another critical component of $O_X$, it exhibited statistically significant positive correlations with not only $SOC_{SOA}$ and $SOC_{PMF,}$
but also several TBM estimated SOCs, including $SOC_{TBM}$, $SOC_{\_Mono}$, $SOC_{\_Nap}$, and $SOC_{\_Cary}$. This may be because SOA tracers
used in TBM were produced from the photo-oxidation of these VOC precursors in the presence of NOx (Kleindienst et al.,
2007). The significant positive correlations between $NO_2$ and $SOC_{SOA}$ and $SOC_{PMF}$ also suggests that the daytime oxidation
processes involving $NO_X$ are critical SOA formation pathways in the region. Significant correlations with R>0.5 between $NO_3$
and $SOC_{\_Mono}$, $SOC_{SOA}$, $SOC_{PMF}$, and $SOC_{SS}$ were also observed. BVOCs were found to account for >80% of the $NO_3$ reactivity
in Hong Kong (Brown et al., 2016), with monoterpenes as the leading contributor. Both Zhang et al. (2018) and Xu et al. (2013)
have reported an enhancement of nighttime monoterpenes SOA in the southeastern U.S. by $NO_3$-monoterpenes reactions.
Therefore, our findings indicate that SOA formation through nighttime $NO_3$ oxidation of biogenic VOCs, especially
monoterpenes, may have made a considerable contribution to the SOA loading in Hong Kong. Since $NO_3$ is a key precursor
of nighttime production of $HNO_3$, and nitrate is a significant component of secondary inorganic aerosols, it rationalized the
correlations between $NO_3$ and $SOC_{SS}$. Six SOC terms, i.e., $SOC_{\_Mono}$, $SOC_{\_Nap}$, $SOC_{TBM}$, $SOC_{SOA}$, $SOC_{SS}$, and $SOC_{PMF}$, showed
significant positive correlations with sulfate, especially $SOC_{SS}$ and $SOC_{PMF}$ (R≥0.8, P<0.01). Given that sulfate is the key
component of secondary inorganic aerosol, such a strong correlation between $SOC_{SS}$ and sulfate is expected. Moreover, several
studies have suggested that sulfate also plays a dominant role in the production of aerosol-phase organosulfates through both
nucleophilic addition reactions and the salting-in effect (Lin et al., 2012; Riva et al., 2015; Xu et al., 2015).
We then performed multivariate linear regression (MLR) analysis to obtain a quantitative and comprehensive understanding
of the impacts of gaseous oxidants and aerosol characteristics on $SOC_{TBM}$, $SOC_{PMF}$, and the individual PMF resolved SOCs
(i.e., $SOC_{SS}$, $SOC_{SOA}$, and $SOC\_{BB}$). Six parameters, namely $O_3$, $NO_2$, $NO_3$, sulfate, $H_P^+$, and $LWC_P$, were included in the
preliminary runs. However, the MLR results showed that $O_3$ was an insignificant factor for all SOC terms, even with negative
regression coefficients. Pearson's R analysis also showed that SOC was more $NO_2$ dependent than $O_3$. Therefore, it was
excluded from the final MLR analysis, and the results were shown in Table 4.
We found that three parameters, i.e., $NO_2$, sulfate, and $NO_3$, have statistically significant positive linear relationships ($P \leq$
0.001) with $SOC_{SS}$, and the regression coefficients were 0.303, 0.913, and 0.234, respectively. The result is reasonable and
consistent with what was observed from the Pearson's R analysis, given that sulfate is the critical component in the PMF
resolved SS factor, and both $NO_2$ and $NO_3$ are the precursors of nitrate through $HNO_3$ formation. As for $SOC\_{BB}$, three
parameters, i.e., $NO_2$, $NO_3$, and $H_P^+$, showed significant positive linear relationships with it (P<0.01), with a regression
coefficient of 0.639, 0.509, and 0.503, respectively. This indicates that 1 mol $L^{-1}$ increase in particle acidity was associated
with a 0.503 µgC $m^{-3}$ increase in SOC from BB aging. Phenols, which are produced from the combustion of lignin, are a typical
class of gaseous compounds emitted in large amounts from BB (Bruns et al., 2016; Schauer et al., 2001). Recent laboratory
studies indicate that phenols can undergo multiphase photochemical reactions in the atmosphere with the formation of
nitrophenols and nitrocatechols (Finewax et al., 2018; Yu et al., 2014). Vione et al. (2001) observed the aqueous phase
photonitration of phenols, which was pH-dependent with more nitro-compounds generated at lower pH. Given the strong
particle acidity (pH annual mean: -0.28) observed in the Hong Kong atmosphere, the formation of the 4-nitrocatechol and its
analogs may be favored in the BB plume, which enhances $SOC\_{BB}$ formation.
Both sulfate and $NO_2$ were found as the statistically significant factors that positively correlated with $SOC_{PMF}$, with
regression coefficients of 0.53 and 0.37, respectively (P<0.001, Table 4). This suggests reducing the sulfate level by 1 µg $m^{-3}$
and $NO_2$ level by 1 ppb could lower the total PMF-apportioned SOC by 0.53 and 0.37 µgC $m^{-3}$, respectively. $NO_2$ was also
the most significant factor influencing $SOC_{TBM}$, with a regression coefficient of 0.38 (P<0.001), indicating that a decrease of
$NO_2$ by 1 ppb can reduce $SOC_{TBM}$ by 0.38 µgC $m^{-3}$. As for $SOC_{SOA}$, we found $NO_3$ as the most significant parameter (P<0.01),
and a decrease of 1 ppb $NO_3$ can lead to a reduction of $SOC_{SOA}$ by 0.38 µgC $m^{-3}$ when holding other covariates unchanged.
These results are consistent with what was observed from the Pearson's R analysis, indicating the importance of NOx
processing on both daytime and nighttime SOA production in the region.
**4 Conclusions**
In this study, we identified and quantitatively assessed the contributions of six primary and secondary sources to ambient
OC in Hong Kong, and secondary formation was found to be the leading contributor. Anthropogenic emissions, including $NO_2$,



$O_X$, $NO_3$, and sulfate, significantly influenced SOA formation in the region. In particular, NOx processing in both daytime and
nighttime has played a critical role. Although the ambient $NO_2$ level has dropped by 33.3% from 1999 to 2019 (the government
of HKSAR, https://www.info.gov.hk/gia/general/202001/20/P2020012000874.htm) and sulfate level in $PM_{2.5}$ was also
lowered by about 30% from 2000 to 2016 (HKEPD, 2017), the roadside $NO_2$ level was still high. According to the 20-year air
pollutants monitoring data released by HKSAR, the annual average concentration of roadside $NO_2$ was much higher than the
other gaseous pollutants, and it peaked during 2011-2013, which were 122 and 118 µg m$^{-3}$ in 2011 and 2012, respectively.
Although the annual ambient level of roadside $NO_2$ decreased to 80 µg m$^{-3}$ in 2019, it is still two times higher than the annual
objective level set by the HKSAR government, indicating a continuous significant impact of $NO_X$ on SOA formation in Hong
Kong, especially in areas with heavy traffic load. Given that 90% of the roadside $NO_2$ was from commercial vehicles, such as
buses, trucks, minibuses, and so on, our results suggest that more stringent control of $NO_X$ emission from commercial vehicles
is needed. This will benefit the community by reducing not only the background $NO_X$ levels but also the SOA pollution in
Hong Kong.
**Appendices**
**Appendix A: Kinetic model of loss of isoprene intermediates**
In this study, we use Kintecus, a kinetics simulation software, to investigate the degradation pathways of two isoprene SOA
intermediates, i.e., IEPOX and MAE, in the atmosphere. Simulation time was set to be 100 h to ensure the completion of
reactions. As described by Eddingsass et al. (2010) and Worton et al. (2013), IEPOX and MAE are removed from the
atmosphere mainly through three pathways, namely the gas-phase photo-oxidation, dry deposition, and aerosol phase acid-
catalyzed ring-opening reaction. Reaction constants that are involved in these three degradation processes were listed below.

IEPOX:

$$k_{OX} = 5.78 \times 10^{-11} \cdot e^{-400/T} \cdot [OH] \ s^{-1}$$

$$k_{dd} = dv/blh \ s^{-1}$$

$$k_{H^+} = 5 \times 10^{-2} \cdot [H_P^+] \ s^{-1}$$

$$k_H^{cp} = 1.3 \times 10^8 \ M \ atm^{-1}$$

MAE:

$$k'_{OX} = 1.0 \times 10^{-12} \cdot [OH] \ s^{-1}$$

$$k'_{dd} = dv/blh \ s^{-1}$$

$$k'_{H^+} = 5.91 \times 10^{-5} \cdot [H_P^+] \ s^{-1}$$

$$k'^{cp}_H = 7.5 \times 10^6 \ M \ atm^{-1}$$

The eight terms, i.e., $k_{OX}$ and $k'_{OX}$, $k_{dd}$ and $k'_{dd}$, $k_{H^+}$ and $k'_{H^+}$, and $K_H^{cp}$ and $K'^{cp}_H$, are the gas-phase oxidation rate
constants, dry deposition rate constants, acid-catalyzed ring-opening rate constants, and Henry's law constants of IEPOX and
MAE, respectively. Given the annual average OH radical level in the PRD region was $5 \times 10^6$ molecules cm$^{-3}$ (Hofzumahaus et
al., 2009), $k_{OX}$ and $k'_{OX}$ were calculated to be $7.55 \times 10^{-5}$ s$^{-1}$ and $5.12 \times 10^{-6}$ s$^{-1}$ at 298 K. $k_{dd}$ is estimated by the deposition
velocity (dv) and the boundary layer height (blh). Like Eddingsaas et al. (2010) and Worton et al. (2013), we assumed the same





deposition velocities for IEPOX and MAE as that for hydrogen peroxide (1-5 cm s$^{-1}$). With the predicted boundary height in
Hong Kong of 1100 m (Xie et al., 2012), $k_{dd}$ and $k_{dd}'$ were calculated to be $5.05\times10^{-5}$ s$^{-1}$. Given the high volatility of MAE
vapor pressure ($9.2\times10^{-5}$ atm) (Worton et al., 2013), it has a low tendency to partition onto the particle phase and its uptake
onto aqueous particles is mainly governed by Henry's law constant ($k_H{}^{cp}$). Worton et al. (2013) estimated the $k_H{}^{cp}$ value of
MAE to be $7.5\times10^6$ M atm$^{-1}$, which is 20 times lower than that of IEPOX ($1.3\times10^8$ M atm$^{-1}$, Minerath et al., 2008). Moreover,
Riedel et al. (2015) suggested that the heterogeneous reactive uptake coefficient of MAE ($\gamma = 4.9\times10^{-4}$) through the ring-
opening reaction was a factor of 30 lower than that of IEPOX. The ring-opening rate constant ($k_{H+}$) for IEPOX and MAE were
estimated by Eddingsaas et al. (2010) and Birdsall et al. (2014), which are $5\times10^{-2}$ M$^{-1}$ s$^{-1}$ and $5.91\times10^{-5}$ M$^{-1}$ s$^{-1}$, respectively.
We then inputted all these parameters into the Kintecus model and estimated the fractions of IEPOX and MAE degraded
through the above mentioned three pathways.
**Appendix B: Calculation of particle acidity and total liquid water content**

A thermodynamic *model* (*E-AIM model II*) was applied to estimate the hydrogen ion concentration in air (H$^+_{air}$) and liquid

water content associated with inorganic species (LWC$_{inorg}$). The liquid water content associated with organic species (LWC$_{org}$)
was calculated using the following equation
$$LWC_{org} = \frac{m_{org}\rho_w}{\rho_{org}}\frac{k_{org}}{(1/RH - 1)}$$

where $k_{org}$ is an organic hygroscopicity parameter and has a value of 0.1, $m_{org}$ is organic mass concentration, and a factor

of 2.1 was applied to convert OC to OM at the urban location. $\rho_w$ is the water density, and a typical value of 1.4 g cm$^{-1}$ was
applied for organic aerosols ($\rho_{org}$). Since LWC is associated with both inorganic and organic species, the total particle water
(LWC$_p$) was calculated as the sum of LWC$_{inorg}$ and LWC$_{org}$ based on the assumption that particles were internally well mixed.

Particle acidity was calculated using the following equation:

$$H_p^+ = \frac{1000H_{air}^+}{LWC_{org} + LWC_{inorg}}$$

where $H_p^+$ (mol L$^{-1}$) is the concentration of hydrogen ion in aerosol water, interpreted as particle acidity. $H_{air}^+$ and LWC$_{inorg}$
were calculated by E-AIM model II using input values of inorganic ions, RH, and temperature.
**Data availability.** Raw data used in this study are archived at Hong Kong Baptist University, and are available upon request
by contacting the corresponding author.
**Author contributions.** YBC and DH designed the study. YBC did all the experiments and most of the data analysis. YQM
helped model analysis and data interpretation. YBC drafted the manuscript. DH helped with data analysis and interpretation
and revised the manuscript.
**Competing interest.** The authors declare that they have no conflict of interest.




**Acknowledgment**

This work was supported by the National Natural Science Foundation of China (21976151 and 21477102) and the General
Research Fund of Hong Kong Research Grant Council (12328216, 12304215, 12300914). The authors thank the
Environmental Central Facility (ENVF) in Hong Kong University of Science and Technology (HKUST) for real-time
environmental and air quality data (http://envf.ust.hk/dataview/gts/current/).



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





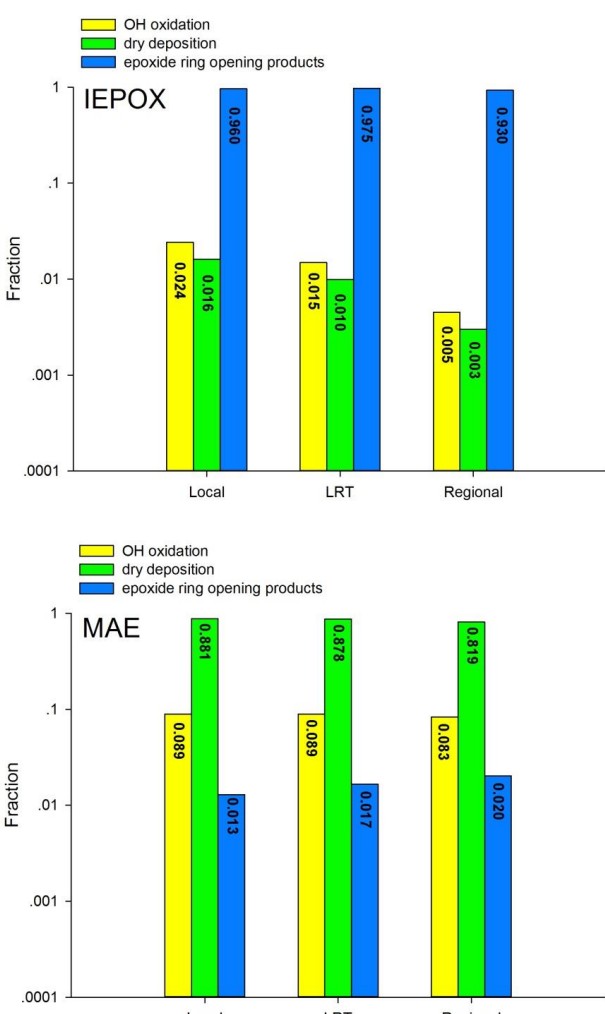


Figure 1: Comparison of three degradation processes for IEPOX and MAE under the three synoptic conditions



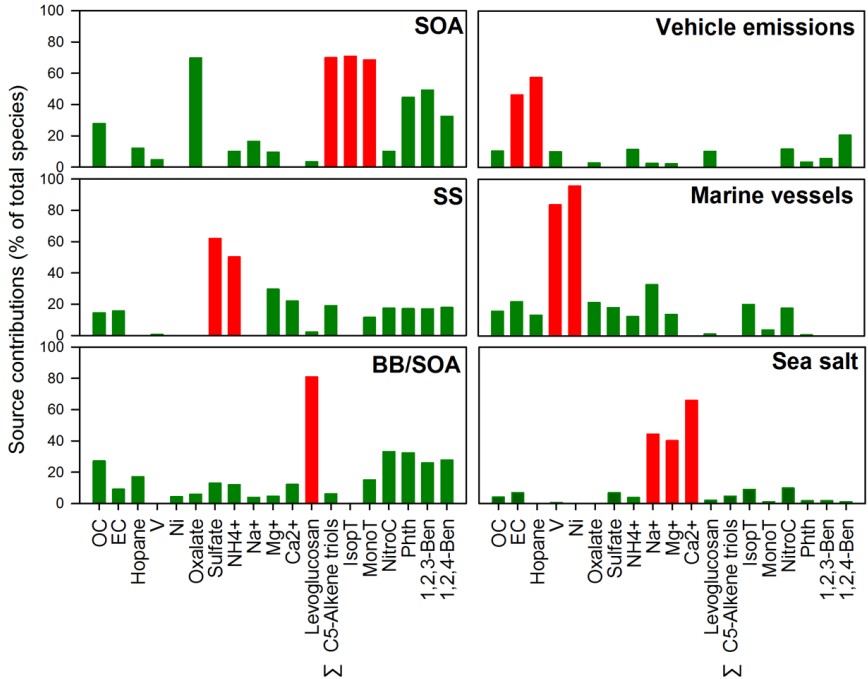


Figure 2: PMF-resolved source contributions (% of total species) to ambient $PM_{2.5}$ samples collected in Hong Kong. Red
column: chemical markers for source identification.

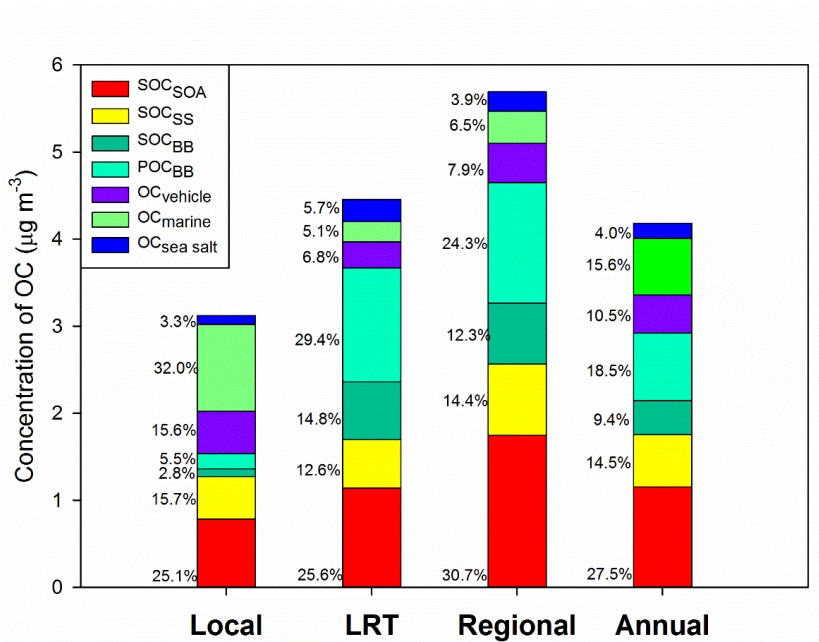


Figure 3: Source-specific contributions to OC under different meteorological conditions. $OC_{BB}$ was split into $POC_{BB}$ and
$SOC_{BB}$.

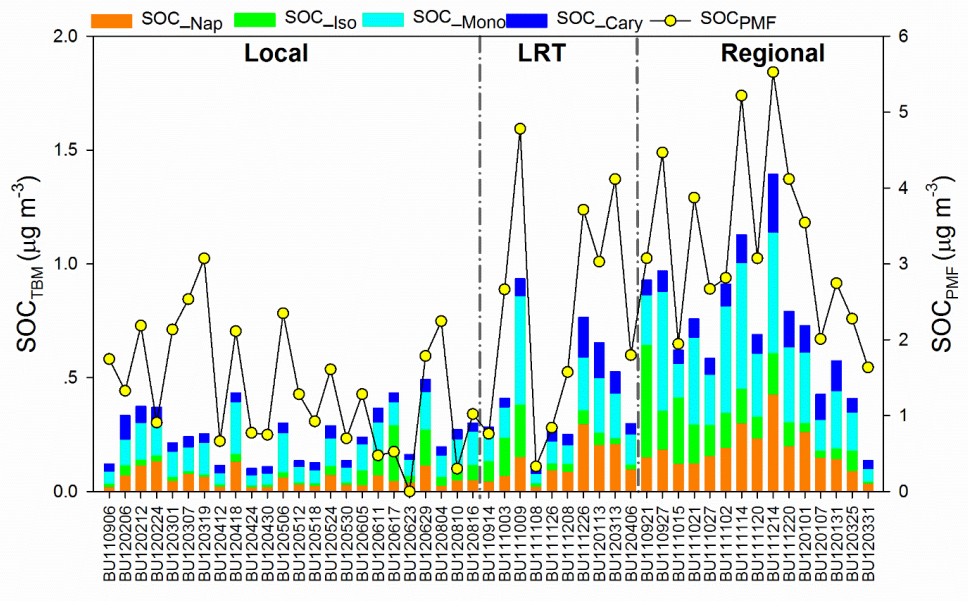


Figure 4: Temporal variations of $SOC_{PMF}$ and $SOC_{TBM}$.


Table 1: Concentrations of 15 SOA tracers, 25 polar organic compounds, and nine inorganic ions in PM$_{2.5}$ collected in Hong
Kong under three meteorological conditions.

| | Local (N=24) | | Long regional transport (N=10) | | Regional (N=15) | |
|---|---|---|---|---|---|---|
| | Average | Range | Average | Range | Average | Range |
| *Tracers for isoprene SOA (ng m$^{-3}$)* | | | | | | |
| 2-Methylglyceric acid | 0.56±0.31 | 0.22-1.42 | 1.28±0.86 | 0.29-2.61 | 2.36±1.75 | 0.02-6.42 |
| 2-Methylthreitol | 2.34±3.95 | 0.33-18.79 | 2.88±2.97 | 0.57-9.60 | 6.23±5.69 | 0.35-23.37 |
| 2-Methylerythritol | 7.06±13.95 | 0.54-64.67 | 6.94±7.81 | 1.08-23.58 | 13.49±12.23 | 0.48-47.62 |
| cis-2-Methyl-1,3,4-trihydroxy-1-butene | 0.87±1.25 | 0.15-6.06 | 2.00±2.56 | 0.33-8.62 | 5.78±4.57 | 0.22-17.19 |
| 3-Methyl-2,3,4-trihydorxy-1-butene | 0.52±0.49 | 0.15-2.00 | 1.03±1.17 | 0.23-4.08 | 2.40±1.91 | 0.18-7.31 |
| trans-2-Methyl-1,3,4-trihydorxy-1-butene | 1.28±1.25 | 0.15-5.08 | 5.25±5.59 | 0.48-18.68 | 10.52±8.19 | 0.37-25.23 |
| 3-MeTHF-3,4-diols | 0.18±0.06 | 0.15-0.34 | 0.21±0.07 | 0.15-0.32 | 0.29±0.12 | 0.15-0.60 |
| ∑C5-Alkene triols | 2.68±2.52 | 0.45-8.89 | 8.27±8.92 | 1.20-31.37 | 18.71±13.14 | 0.78-40.08 |
| ∑Isoprene tracers (exclude triols) | 10.06±18.09 | 1.22-84.87 | 11.23±11.18 | 1.93-35.70 | 22.32±18.94 | 1.17-77.09 |
| ∑Isoprene tracers | 12.74±20.13 | 1.67-93.41 | 19.51±19.41 | 3.14-67.07 | 41.03±29.71 | 1.95-117.17 |
| *Tracers for monoterpenes SOA (ng m$^{-3}$)* | | | | | | |
| 3-Hydroxyglutaric acid | 3.40±2.09 | 0.72-9.14 | 6.11±5.42 | 0.66-19.15 | 11.53±6.27 | 1.35-22.04 |
| 3-Hydroxy-4,4-dimethylglutaric acid | 0.53±0.12 | 0.42-0.93 | 0.71±0.30 | 0.43-1.39 | 0.91±0.28 | 0.41-1.39 |
| 3-Methyl-1,2,3-butanetricarboxylic acid | 0.59±0.19 | 0.40-1.18 | 0.84±0.39 | 0.45-1.76 | 1.28±0.52 | 0.42-2.14 |
| 3-Isopropylpentanedioic acid | 1.07±0.38 | 0.55-1.85 | 1.52±0.86 | 0.51-3.46 | 2.57±1.52 | 0.61-4.86 |
| 3-Acetyl pentanedioic acid | 0.82±0.23 | 0.45-1.19 | 1.13±0.55 | 0.49-2.42 | 1.71±0.87 | 0.54-3.20 |
| ∑ Monoterpenes tracers | 6.41±2.75 | 2.63-13.49 | 10.31±7.33 | 2.54-28.17 | 18.00±9.28 | 3.33-32.57 |
| *Tracers for β-caryophyllene SOA (ng m$^{-3}$)* | | | | | | |
| β-Caryophyllinic acid | 0.94±0.41 | 0.49-2.36 | 1.73±1.16 | 0.75-3.99 | 2.33±1.21 | 0.80-5.82 |
| *Tracers for Naphthalene SOA (ng m$^{-3}$)* | | | | | | |
| Phthalic acid | 2.26±1.38 | 0.80-5.17 | 4.97±3.30 | 0.92-11.41 | 7.16±3.61 | 1.41-16.42 |
| *Dicarboxylic acids (ng m$^{-3}$)* | | | | | | |
| Succinic acid | 2.10±1.63 | 0.65-6.23 | 4.56±4.80 | 0.80-14.18 | 5.27±3.43 | 0.68-12.19 |
| Maleic acid | 0.42±0.27 | 0.14-1.47 | 0.42±0.23 | 0.14-0.84 | 0.36±0.18 | 0.15-0.78 |
| Malic acid | 2.67±1.49 | 0.64-5.59 | 4.20±3.74 | 0.60-13.12 | 8.10±4.12 | 1.33-13.86 |
| Glutaric acid | 2.63±6.06 | 0.82-30.89 | 2.36±1.73 | 0.79-4.99 | 2.85±1.53 | 0.67-6.05 |
| Citramalic acid | 0.76±0.23 | 0.38-1.30 | 0.86±0.32 | 0.38-1.48 | 1.23±0.47 | 0.52-2.00 |
| Terephthalic acid | 9.28±7.49 | 2.16-31.86 | 30.21±27.20 | 3.58-79.61 | 36.89±23.84 | 3.77-79.25 |
| Adipic acid | 1.34±1.42 | 0.54-6.20 | 1.20±0.46 | 0.64-2.21 | 1.48±0.66 | 0.67-3.08 |
| Pimelic acid | 0.68±0.10 | 0.51-0.93 | 0.82±0.29 | 0.52-1.47 | 0.99±0.35 | 0.52-1.94 |
| Oxalic acid (μg m$^{-3}$) | 0.35±0.20 | 0.11-0.86 | 0.38±0.23 | 0.09-0.72 | 0.54±0.21 | 0.29-0.94 |
| *Saccharides (ng m$^{-3}$)* | | | | | | |
| Levoglucosan | 22.51±41.16 | 0.64-161.16 | 120.79±129.55 | 3.21-362.74 | 128.52±140.39 | 8.64-474.15 |
| Meso-erythritol | 0.11±0.10 | 0.03-0.43 | 0.29±0.25 | 0.03-0.74 | 0.44±0.28 | 0.07-1.22 |
| Xylitol | 0.29±0.11 | 0.21-0.69 | 0.50±0.28 | 0.23-1.02 | 0.52±0.22 | 0.22-1.03 |
| Xylose | 1.24±1.08 | 0.50-4.57 | 4.65±4.45 | 0.58-13.34 | 5.34±4.31 | 0.81-16.12 |
| Galactose | 1.82±2.02 | 0.37-9.97 | 3.31±1.97 | 1.09-7.08 | 3.51±1.71 | 1.02-6.84 |
| Mannitol | 0.16±0.04 | 0.12-0.26 | 0.21±0.07 | 0.13-0.37 | 0.23±0.07 | 0.13-0.37 |
| Fructose | 2.30±3.19 | 0.26-15.58 | 3.64±3.89 | 0.38-13.41 | 4.32±2.54 | 1.65-9.32 |
| Galactosan | 1.09±0.53 | 0.79-2.99 | 2.58±2.47 | 0.84-7.20 | 2.68±2.40 | 0.88-7.99 |
| Sorbitol | 1.45±0.37 | 1.14-2.54 | 1.55±0.28 | 1.21-1.96 | 1.70±0.40 | 1.31-2.62 |
| Glucose | 1.55±0.89 | 0.50-3.83 | 1.20±0.61 | 0.40-2.07 | 1.51±0.92 | 0.52-3.29 |
| Sucrose | 0.94±1.81 | 0.42-9.43 | 0.58±0.14 | 0.42-0.91 | 0.57±0.08 | 0.45-0.76 |
| Arbitol | 0.25±0.10 | 0.00-0.57 | 0.40±0.20 | 0.22-0.78 | 0.42±0.17 | 0.22-0.85 |
| *Other compounds (ng m$^{-3}$)* | | | | | | |



| | | | | | | |
|---|---|---|---|---|---|---|
| 4-Nitrocatechol | 0.90±0.12 | 0.78-1.35 | 1.30±0.62 | 0.84-2.75 | 1.55±0.83 | 0.85-4.00 |
| Cholesterol | 1.29±0.25 | 0.94-1.81 | 1.30±0.28 | 1.01-1.93 | 1.20±0.27 | 0.95-1.89 |
| 1,2,3-Benzenetricarboxylic Acid | 1.23±0.67 | 0.47-2.46 | 2.25±1.34 | 0.63-4.70 | 3.97±2.54 | 0.54-9.50 |
| 1,2,4-Benzenetricarboxylic Acid | 1.77±1.28 | 0.47-6.17 | 3.32±2.34 | 0.88-6.77 | 5.16±3.30 | 0.73-12.54 |
| | | | *Major ion ($\mu g\ m^{-3}$)* | | | |
| Sulfate | 11.43±5.98 | 3.28-30.32 | 13.02±9.25 | 1.49-29.25 | 17.35±5.20 | 8.90-29.29 |
| Nitrate | 0.89±1.17 | 0.05-3.39 | 1.62±2.10 | 0.08-5.84 | 1.41±1.51 | 0.38-5.49 |
| Chloride | 0.18±0.17 | 0.06-0.77 | 0.17±0.15 | 0.07-0.45 | 0.14±0.09 | 0.07-0.40 |
| Ammonia | 2.05±0.91 | 0.47-4.12 | 2.26±1.48 | 0.30-4.36 | 2.99±0.72 | 1.82-4.69 |
| Potassium | 0.11±0.07 | 0.03-0.36 | 0.29±0.17 | 0.05-0.49 | 0.40±0.22 | 0.15-0.94 |
| Magnesium | 0.01±0.01 | 0.00-0.03 | 0.02±0.01 | 0.00-0.04 | 0.02±0.01 | 0.00-0.04 |
| Calcium | 0.03±0.03 | 0.00-0.13 | 0.08±0.07 | 0.02-0.23 | 0.08±0.04 | 0.02-0.15 |
| Sodium | 0.09±0.09 | 0.01-0.40 | 0.16±0.14 | 0.03-0.52 | 0.14±0.06 | 0.08-0.30 |






Table 2: PMF and TBM-resolved OCs, concentrations of gas pollutants, PM$_{2.5}$, EC, OC, and major aerosol characteristics
under different meteorological conditions.

| | Local (N=24) | | Long regional transport (N=10) | | Regional (N=15) | | Annual (N=49) | |
|---|---|---|---|---|---|---|---|---|
| | Average | Range | Average | Range | Average | Range | Average | Range |
| PM$_{2.5}$ (µg m$^{-3}$) | 24.11±9.99 | 10.04-49.28 | 32.23±14.81 | 7.63-50.68 | 38.5±10.48 | 26.20-65.28 | 30.17±12.72 | 7.63-65.28 |
| EC (µgC m$^{-3}$) | 1.02±0.57 | 0.47-2.75 | 0.85±0.60 | 0.14-2.10 | 1.14±0.45 | 0.50-2.12 | 1.02±0.54 | 0.14-2.75 |
| OC$_{measured}$ | 2.94±1.11 | 1.61-5.75 | 4.16±2.53 | 1.25-8.53 | 6.15±2.51 | 3.21-12.97 | 4.18±2.37 | 1.25-12.97 |
| *PMF apportioned OC (µgC m$^{-3}$)* | | | | | | | | |
| SOC$_{SOA}$ | 0.78±0.65 | 0.00-2.27 | 1.14±0.82 | 0.18-2.72 | 1.75±0.75 | 0.65-3.29 | 1.15±0.82 | 0.00-3.29 |
| SOC$_{SS}$ | 0.49±0.37 | 0.00-1.74 | 0.56±0.67 | 0.00-1.81 | 0.82±0.38 | 0.24-1.65 | 0.60±0.46 | 0.00-1.81 |
| OC$_{BB}$ (POC$_{BB}$+SOC$_{BB}$) | 0.26±0.63 | 0.00-2.34 | 1.97±2.26 | 0.00-6.34 | 2.08±2.63 | 0.00-8.96 | 1.17±1.99 | 0.00-8.96 |
| OC$_{Vehicle}$ | 0.49±0.46 | 0.00-2.07 | 0.30±0.42 | 0.00-1.26 | 0.45±0.36 | 0.01-1.26 | 0.44±0.42 | 0.00-2.07 |
| OC$_{Marine}$ | 1.00±0.63 | 0.04-2.97 | 0.23±0.19 | 0.00-0.51 | 0.37±0.21 | 0.08-0.71 | 0.65±0.18 | 0.00-2.97 |
| OC$_{Sea\ salt}$ | 0.10±0.11 | 0.00-0.53 | 0.25±0.33 | 0.00-1.13 | 0.22±0.16 | 0.00-0.62 | 0.17±0.19 | 0.00-1.13 |
| SOC$_{BB}$ | 0.09±0.21 | 0.00-0.79 | 0.66±0.76 | 0.00-2.13 | 0.70±0.88 | 0.00-3.01 | 0.39±0.67 | 0.00-3.01 |
| SOC$_{PMF}$ | 1.36±0.81 | 0.00-3.07 | 2.36±1.54 | 0.33-4.78 | 3.27±1.18 | 1.63-5.53 | 2.15±1.37 | 0.00-5.53 |
| SOC$_{PMF}$/OC (%) | 43.0±16.8% | 0.0%-66.5% | 52.3±21.1% | 30.0%-85.3% | 60.2±13.7% | 36.2%-78.8% | 50.2±18.2% | 0.0%-85.3% |
| *Tracer based method estimated OC (µgC m$^{-3}$)* | | | | | | | | |
| SOC$_{\_Iso}$ | 0.04±0.06 | 0.01-0.24 | 0.07±0.07 | 0.01-0.23 | 0.14±0.12 | 0.01-0.49 | 0.08±0.09 | 0.01-0.49 |
| SOC$_{\_Mono}$ | 0.14±0.06 | 0.06-0.29 | 0.22±0.16 | 0.05-0.60 | 0.38±0.20 | 0.07-0.69 | 0.23±0.17 | 0.05-0.69 |
| SOC$_{\_Cary}$ | 0.04±0.02 | 0.02-0.10 | 0.08±0.05 | 0.03-0.17 | 0.10±0.05 | 0.03-0.25 | 0.07±0.05 | 0.02-0.25 |
| SOC$_{\_Nap}$ | 0.06±0.04 | 0.02-0.13 | 0.13±0.09 | 0.02-0.30 | 0.19±0.09 | 0.04-0.43 | 0.11±0.09 | 0.02-0.43 |
| SOC$_{TBM}$ | 0.28±0.13 | 0.11-0.53 | 0.50±0.29 | 0.12-1.06 | 0.81±0.35 | 0.15-1.53 | 0.49±0.34 | 0.11-1.53 |
| SOC$_{TBM}$/OC | 10.2±5.1% | 3.8%-22.7% | 13.0±4.6% | 5.3%-20.7% | 13.4±4.3% | 4.7%-19.6% | 11.8±4.9% | 3.8%-22.7% |
| *Gas Pollutants and other aerosol characteristics* | | | | | | | | |
| O$_{3\_average}$ (ppb) | 11.61±7.3 | 2.93-32.12 | 13.96±7.94 | 2.86-26.92 | 20.64±8.74 | 2.88-31.84 | 14.85±8.69 | 2.86-32.12 |
| NO$_{2\_average}$ (ppb) | 34.56±10.66 | 16.7-54.32 | 34.59±7.62 | 21.74-42.85 | 42.98±7.10 | 32.72-60.37 | 37.15±9.76 | 16.70-60.37 |
| SO$_{2\_average}$ (µg m$^{-3}$) | 4.14±2.92 | 0.7-10.38 | 3.81±1.88 | 2.23-7.30 | 5.38±2.24 | 2.96-10.45 | 4.45±2.57 | 0.70-10.45 |
| O$_X$ (µg m$^{-3}$) | 87.45±26.26 | 49.72-138.49 | 93.18±21.37 | 61.66-125.79 | 122.39±17.70 | 69.54-145.90 | 99.31±27.42 | 49.72-145.90 |
| p[NO3] (ppb h$^{-1}$) | 1.25±0.96 | 0.30-4.17 | 1.36±0.94 | 0.31-3.29 | 2.45±1.02 | 0.23-3.76 | 1.64±1.10 | 0.23-4.17 |
| NO$_{3\_average}$ (ppb) | 0.05±0.04 | 0.01-0.18 | 0.06±0.04 | 0.01-0.14 | 0.10±0.04 | 0.01-0.16 | 0.07±0.05 | 0.01-0.18 |
| H$_P^+$ (M) | 1.72±1.04 | 0.02-3.78 | 2.66±1.50 | 0.49-5.43 | 3.22±0.79 | 2.31-4.76 | 2.37±1.25 | 0.02-5.43 |
| pH | (-0.20)±0.52 | (-0.58)-1.81 | (-0.31)±0.32 | (-0.74)-0.31 | (-0.50)±0.10 | (-0.68)-(-0.36) | (-0.28)±0.42 | (-0.74)-1.81 |
| LWC (µg m$^{-3}$) | 66.64±46.51 | 2.68-184.71 | 42.88±28.80 | 6.60-86.03 | 51.65±17.69 | 30.51-101.12 | 57.2±37.15 | 2.68-184.71 |






Table 3: Regression analysis (Pearson's R) of PMF and TBM-resolved SOCs, SO$_2$, NO$_2$, ozone (O$_3$) , particle acidity (H$_P^+$),
total particle-phase liquate water content (LWC$_P$), and sulfate **: P<0.01; *: P<0.05. Note: R>0.5 are bold.

| | Pearson's R | | | | | | | | |
|---|---|---|---|---|---|---|---|---|---|
| | SOC_Iso | SOC_Mono | SOC_Cary | SOC_Nap | SOC_TBM | SOC_BB | SOC_SOA | SOC_SS | SOCPMF |
| O$_3$ (ppb) | 0.374** | .401** | 0.011 | 0.246 | .374** | -0.111 | **.502**** | **.557**** | .434** |
| NO$_2$ (ppb) | .064 | **.516**** | **.586**** | **.528**** | **.500**** | .469** | **.570**** | 0.165 | **.627**** |
| SO$_2$ (ppb) | 0.044 | 0.198 | .463** | .296* | 0.255 | .357* | 0.035 | -0.052 | 0.179 |
| O$_X$ (µg m$^{-3}$) | 0.257 | **.600**** | .433** | **.535**** | **.577**** | 0.281 | **.707**** | .445** | **.711**** |
| NO$_3$ (ppb) | .413** | **.530**** | 0.101 | .313* | .480** | -0.077 | **.637**** | **.574**** | **.538**** |
| Sulfate (µg m$^{-3}$) | .287* | **.610**** | .405** | **.506**** | **.579**** | 0.23 | **.646**** | **.886**** | **.799**** |
| H$_P^+$ (M) | 0.249 | .334* | .391** | .388** | .395** | .400** | 0.164 | 0.24 | .376** |
| LWC$_P$ (µg m$^{-3}$) | -0.18 | 0.18 | 0.115 | 0.209 | 0.113 | 0.003 | .413** | .438** | .397** |


Table 4: Results of multivariate linear analysis of PMF and TBM-resolved SOCs, O$_X$, NO$_3$, sulfate, particle acidity (H$_P^+$), and
total particle-phase liquate water content (LWC$_P$). **: P<0.01; *: P<0.05. Note: significant regressions are bold.

| | normalized β-coefficient | | | | | | | | |
|---|---|---|---|---|---|---|---|---|---|
| | SOCSS | SOCSOA | SOCBB | SOCPMF | SOC_Iso | SOC_Mono | SOC_Cary | SOC_Nap | SOC_TBM |
| O$_X$ (µg m$^{-3}$) | **-0.453**** | 0.355 | **1.08**** | **0.59**** | -0.091 | 0.439* | **1.045**** | **0.739**** | 0.537* |
| NO$_3$ (ppb) | **0.497**** | 0.186 | **-1.159**** | -0.289* | 0.375 | -0.066 | **-0.999**** | -0.519* | -0.204 |
| Sulfate (µg m$^{-3}$) | **0.877**** | 0.334* | 0.164 | **0.576**** | 0.339 | 0.457* | 0.323 | 0.258 | 0.439* |
| H$_P$ (M) | -0.141 | -0.091 | **0.386**** | 0.087 | -0.127 | 0.001 | 0.265 | 0.268 | 0.072 |
| LWC$_P$ (µg m$^{-3}$) | 0.122 | 0.116 | -0.166 | 0.029 | -0.369 | -0.14 | -0.166 | 0.1 | -0.194 |
