# Peer review of "Tracer-based source apportioning of atmospheric organic carbon and"

_Atmospheric Chemistry and Physics, 2021_

## Author Comment (AC2)

**Point-to-point responses to the reviewers' comments on acp-2021-32**

**Reviewer #1**

The authors reported measurement results of ambient particulate matter in Hong Kong using detailed chemical speciation methods. The results of the speciated organic portion by GC-MS suggested dominant contribution of secondary organic carbon (SOC) to total organic carbon (OC) measured. Focusing on the loading and sources of SOC, the authors used positive matrix factorization (PMF) and tracer-based method (TBM) to quantify the concentrations and source contributions of SOC in Hong Kong. Results showed that these two methods tracked well, but the latter method gave a much lower SOC loading compared to the former method (by a factor of 4). The authors went on to use linear correlation and multi-linear correlation to explore the relationships between different SOC types with other air pollutants or indicators, and found that nitrogen oxides (both NOx and NO3) and sulfate might play important roles in SOC formation in Hong Kong.

This is a rigorously performed field measurement study with very detailed chemical speciation information on ambient particulate matter in a typical urban environment in South China. The chemical characterization, data analysis, and discussion are scientifically sound. The manuscript is fairly well written. I therefore recommend Minor Revision with some comments provided below.

Response: Thanks for the reviewer's positive comments.

Major:

1. I was a bit confused about how different sub-types of SOC were calculated. On P11/L304, the authors showed that SOC_BB was calculated from OC_BB. But where was OC_BB from? The "BB/SOA" factor from PMF analysis? In addition, where were SOC_SOA and SOC_SS calculated from? The organics (since mostly polar and believed to be SOA species) in the "SOA" and "SS" factors from the PMF analysis? It would be better to have some description on how the concentrations of those sub-types of SOC were obtained.

   Response: Since SOA and SS are two secondary sources, OC apportioned into these two factors are of secondary origin and were called $SOC_{SOA}$ and $SOC_{SS}$. Yes, $OC\_BB$ is the

amount of OC apportioned into the BB/SOA factor in PMF analysis. As shown in Figure 2 and described in line 292-294 in the revised manuscript, besides levoglucosan, a considerable amount of SOA markers of BB aging, i.e., 4-nitrocatechol, 1,2,3- and 1,2,3-benzenetricarboxylic acids, were apportioned into the BB/SOA factor as well. That's why we considered that OC_BB was not just from the primary emission of BB but also its aging process. We then used equation (1) and the amounts of OC and levoglucosan resolved in the BB/SOA factor to derive the amount of SOC_BB, i.e., OC from BB aging.

In the revised manuscript line 299, we've added, "The amount of OC apportioned to each factor in PMF analysis was considered as the contribution of that source to ambient OC. Therefore, …"

2. Section 3.4: I am not fully convinced that a good correlation between daily concentrations of SOC factors and daily concentration of NO3 can support the statement that oxidation by NO3 radical plays a very important role in SOC formation (although this process might actually be important in Hong Kong, to be confirmed by other evidence). In the daily average concentrations, the diurnal profiles are not reflected, but the most likely case would be that most SOC factors would have peak concentrations during day time, while NO3 would have a peak concentration at night time (although one cannot completely exclude some presence of NO3 at daytime). Therefore, a good correlation between SOC factors and NO3 can at most suggest an indirect relationship between the two.

Response: Since all aerosol phase chemical data reported in this study were 24-h integrated filter-based, we used 24-h averaged levels of gaseous pollutants and environmental parameters when examining the factors that could influence SOA formation. In our study, we have observed significant correlations of $NO_3$ with $SOC_{SOA}$, $SOC_{PMF}$, $SOC\_{Mono}$, and $SOC_{SS}$ in Pearson's R analysis. The MLR analysis also showed that $NO_3$ was the most critical parameter (P<0.01) for $SOC_{SOA}$. Since a total of 49 $PM_{2.5}$ samples taken under different meteorological conditions during the whole year were analyzed, the chemical data have shown considerable concentration variations. We believe the statistically significant relationships between $NO_2$, $NO_3$, and the SOC terms obtained in Pearson's R and MLR analysis indicate that NOx processing during both daytime and nighttime may have played an important role in SOA formation in Hong Kong. A previous study based on field

measurement of nighttime $NO_3$ and $NO_2O_5$ and canister VOCs samples also inferred a significant contribution of biogenic VOCs to $NO_3$ loss in Hong Kong (Brown et al., 2016).

We appreciate the reviewer's comment, and we understand the reviewer's concern about the over-interpretation of data. Therefore, we have deleted the following sentence in the abstract, "" In the revised manuscript, line 427-429, we have also added, "However, more field measurement data, e.g., quantification of the particle-phase organic nitrates using real-time online mass spectrometry techniques, are needed to examine the impact of NOx processing on SOA formation in the region."

Brown, S.S., Dubé, W.P., Tham, Y.J., Zha, Q.Z., Xue, L.K., Poon, S., Wang, Z., Blake, D.R., Tsui, W., Parrish, D.D., Wang, T.: Nighttime chemistry at a high altitude site above Hong Kong, J. Geophys. Res. Atmos., 121, 2457–2475, doi:10.1002/ 2015JD024566, 2016.

Minor:

3. P2/L16: monoterpenes to monoterpene

Response: Revised.

4. P2/L30&31: please add "and" before the last item of a list.

Response: Revised.

5. P8/L203: I do not see any relevance to HO2 channel from the discussion above. The paragraph is mainly about ring-opening reaction of IEPOX under conditions of high LWC and low pH. Am I missing something here?

Response: Yes, this paragraph is mainly about the degradation processes of IEPOX and MAE during the sampling period. Since IEPOX is the intermediate formed from the photo-oxidation of isoprene via $HO_2$ channel under low NOx condition, this pathway is called IEPOX $HO_2$ channel. Same term has showed up in Line 176 and 177 in the manuscript.

6. P10/L269-289: better to put this detailed description of PMF in the Method section.

Response: We have thought about putting the PMF description in the Method section. However, many identified organic compounds were the input species of PMF. Since the results of chemical

characterization were shown in section 3.1, we think keeping the description of PMF in section 3.2 will make it easier for the readers to follow. Moreover, Line 281-289 is about the QA/QC of PMF analysis. Therefore, we decided to keep the PMF description part in section 3.2.

7. P13/L364: just "Offenberg et al. (2017)".

Response: Revised.

8. P14/L390: mass to air mass?

Response: Thanks, it should be air mass, and we have revised it.

9. P15/L410: better use proper scientific notation, instead of engineering notation, for the temperature-dependent rate constant.

Response: Thanks, revised.

**Reviewer #2**

The authors have carried out comprehensive chemical characterization of atmospheric aerosols and investigated the contribution of different sources to aerosol mass under various atmospheric conditions using positive matrix factorization (PMF) and tracer-based method (TBM). This work provide new valuable field data to better understand the sources, formation and composition of atmospheric aerosols in HKSAR. The paper is well written. However, I have one question about the data analysis.

Response: Thanks for the reviewer's positive comments.

Major comment

1. My major question is about the aerosol sample collection and some parts of the data analysis. Line 82, "A high-volume air sampler was used to collect PM2.5 onto a quartz fiber filter (20 cm × 25 cm) at a flow rate of 1.13 m3 min-1 for 24 h.". Could the authors provide more details on how they obtain the 24-h or time averaged values for some parameters (such as NO3, particle acidity (HP+) and particle liquid water content (LWCP)) in their study? Further, the authors have conducted the correlation analysis of SOC terms with O3, NO2, SO2, OX, NO3, sulfate, particle acidity (HP+) and particle liquid water content (LWCP). Given 24-h samples were collected, could

the authors comment how to quantify the effect of NO3 on the formation of atmospheric aerosols using 24-h samples? Same question for some species which have photochemical origins like O3. Could the authors also comment on this?

Response: The liquid water content associated with inorganic species ($LWC_{inorg}$) and hydrogen ion concentration in air ($H^+_{air}$) were calculated using E-AIM model II. E-AIM II is an equilibrium thermodynamic model of the system $H^+$ - $NH_4^+$ - $SO_4^{2-}$ - $NO_3^-$ - $H_2O$, and it is valid from 328 K to <200 K. The species input into the model are $H^+$, $NH_4^+$, $SO_4^{2-}$, and $NO_3^-$, and the input environmental parameters are temperature (K) and relative humidity (RH). Detailed description of E-AIM model is available at http://www.aim.env.uea.ac.uk/aim/aim.php. Since the chemical species were measured using the 24-h $PM_{2.5}$ samples, we can only get their 24-h average concentrations. To be consistent, we also used the daily average values of temperature and RH for model calculation. As a result, the $LWC_{inorg}$ and $H^+_{air}$ values calculated by E-AIM II were their 24-h average levels. We then used the two equations listed in Appendix B to obtain the 24-h average values of $LWC_{org}$, $H^+_p$, and $LWC_p$. As for NO3, it was not monitored at the 18 HKEPD stations and its mixing ratio was estimated using Equation 2. All the aerosol phase chemical data reported in this study were 24-h integrated filter-based. That is why when we examined the factors that could influence SOA formation, 24-h average levels of gas pollutants and environmental parameters were used.

In this study, 49 $PM_{2.5}$ samples were taken during the whole year under different meteorological conditions. As shown in Tables 1 and 2, almost all chemical species, OC, SOC, $PM_{2.5}$, gas pollutants, $H^+_p$, and $LWC_p$ showed stark variations throughout the year. Therefore, the correlation values obtained from Pearson's R analysis should give us an idea about which variables are significant to the SOC terms. Based on Pearson's R results, we picked the significant independent factors and performed MLR analysis to estimate the normalized β-coefficients on these SOC terms. We found that both $NO_2$ and $NO_3$ are statistically significant factors positively correlated with several SOC terms. With these 49 samples of detailed chemical analysis data representing the annual variation of SOC formation in Hong Kong, we believe the statistically significant relationships between $NO_2$, $NO_3$ and the SOC terms obtained in both Pearson's R and MLR analysis indicate that NOx processing during both daytime and nighttime may have played an important role in SOA formation in Hong Kong. A previous study based on field measurement of nighttime $NO_3$ and $NO_2O_5$ and canister VOCs samples also inferred a significant contribution of

biogenic VOCs to NO$_3$ loss in Hong Kong (Brown et al., 2016). Of course, if more real-time field measurement data, e.g., quantification of the particle-phase organic nitrates using online mass spectrometry techniques, are available later on, it can provide more solid evidence to examine the impact of NOx processing on SOA formation in the region.

In the revised manuscript, line 427-429, we have added, "However, more field measurement data, e.g., quantification of the particle-phase organic nitrates using real-time online mass spectrometry techniques, are needed to examine the impact of NOx processing on SOA formation in the region."

Minor comments

2. Line 99, "Air pollutants origin from the northern PRD region can travel together with air masses and transport into Hong Kong. Same as in our previous study (Hu et al., 2010; Ma et al., 2019), we carefully examined the air mass backward trajectories, the spatial distribution patterns of SO2, the concentration levels of both PM2.5 and O3, and the synoptic weather conditions during the sampling period." Please provide this information in the supplement for reference.

Response: PM$_{2.5}$ concentrations at HKBU sampling site during the sampling period are provided in Table 2. The O$_3$ level, the synoptic weather conditions (i.e., wind direction and speed, precipitation, and temperature), and the spatial distribution patterns of SO$_2$ over the 18 Hong Kong air quality monitoring stations on each sampling day can be found at http://envf.ust.hk/dataview/gts/current/. Since HKBU is not among these 18 sites, we then used the real-time environmental and air quality data at Sham Shui Po, the site that is closest to HKBU, as the reference. We have updated the information in the revised manuscript. Please check our response to the following comment.

3. Line 123, "We then categorized all sampling days into three groups, i.e., days mainly influenced by the regional pollution from the PRD region (regional days), days influenced by long-regional transport of air mass from the northern and eastern China (LRT days), and days dominated by the locally generated pollutants (local days)." Could the authors provide the guidelines or justifications for this classification.

Response: The air quality monitoring network in Hong Kong comprises 18 fixed monitoring stations, including 15 general stations and three roadside stations. We classified the sampling days

into three groups mainly based on the back-trajectories of air masses on each sampling day. To assist the classification, we also checked the concentration levels of $O_3$ and $PM_{2.5}$ and the spatial distributions of $SO_2$ over the 18 air quality monitoring stations in Hong Kong (http://envf.ust.hk/dataview/gts/current/). For example, on regional days, higher concentrations of $SO_2$ were normally observed on North and Yuen Long stations than those general stations located in the south area of Hong Kong. We have already published two papers on the characterization of HULIS and water-soluble $PM_{2.5}$-induced oxidative potential using the same set of $PM_{2.5}$ samples (Ma et al., 2019; Cheng et al., 2021), and details on the air mass backward-trajectories under the three sampling categories and the detailed classification of sampling days were already provided in Figure S2 and Table S1 in Ma et al., 2019. The X-axis labels of Figure 4 in this manuscript also showed the classification of the sampling days under the three meteorological conditions.

In the revised manuscript, line 119-127, it has been updated: "As described in our previous studies on the analysis of HULIS and water-soluble $PM_{2.5}$-induced oxidative potential using the same set of $PM_{2.5}$ samples (Ma et al., 2019; Cheng et al., 2021), we carefully examined the air mass backward trajectories and categorized all sampling days into three groups, i.e., days mainly influenced by the regional pollution from the PRD region (regional days), days influenced by long-regional transport of air mass from the northern and eastern China (LRT days), and days dominated by the locally generated pollutants (local days). The concentration levels of both $PM_{2.5}$ and $O_3$, and the spatial distribution patterns of $SO_2$ over the 18 Hong Kong air quality monitoring stations (http://envf.ust.hk/dataview/gts/current/) on each sampling day were also checked to assist the classification. A summary of the classification of sampling days and the typical air mass backward-trajectories under the three meteorological categories were presented in Table S2 and Figure S2 in Ma et al. (2019), respectively."

Ma, Y., Cheng, Y., Qiu, X., Cao, G., Kuang, B., Yu, J.Z., Hu, D.: Optical properties, source apportionment and redox activity of humic-like substances (HULIS) in airborne fine particulates in Hong Kong, Environ. Pollut., 255, 113087, https://doi.org/10.1016/j.envpol.2019.113087, 2019.

Cheng, Y., Ma, Y., Dong, B., Qiu, X., Hu, D.: Pollutants from primary sources dominate the oxidative potential of water-soluble $PM_{2.5}$ in Hong Kong in terms of dithiothreitol (DTT) consumption and hydroxyl radical production, J. Hazard. Mater., 405, 124218, doi:10.1016/j.jhazmat.2020.124218, 2021.

[Figure]

Figure 1. The eighteen air monitoring stations in Hong Kong

https://www.aqhi.gov.hk/en/monitoring-network/air-quality-monitoring-network.html

4. Line 180, "We applied the Kintecus kinetic model to quantitatively evaluate the fractions of these two Isop_SOA intermediates that undergo gas-phase oxidation, aerosol-phase acid-catalyzed ring-opening reaction, and dry deposition processes. Details of the model calculations were provided in the appendices." Do the authors run the model with the 24-h averaged values? With time revised data (if available), how would the simulated results vary with the hour of the days and day of the years (or different seasons) in this study?

Response: The main objective of using Kintecus kinetic model is to provide an overall evaluation of the relative importance of the three removal pathways of IEPOX and MAE (i.e., gas-phase photo-oxidation, dry deposition, and aerosol phase acid-catalyzed ring-opening reaction).

The equations used in Kintecus kinetic model are shown in Appendix A. Since we do not have the real-time measurement data of OH radical, [OH] used for the calculation of gas-phase oxidation ($k_{ox}$) is the annual average OH radical level in the PRD region, which is $5 \times 10^6$ molecules cm$^{-3}$ (Hofzumahaus et al., 2009). As for the dry-deposition rate ($k_{dd}$) of both IEPOX and MAE, a predicted boundary height of 1100 m in Hong Kong was used. So no 24-h averaged measurement values were used for the calculation of $k_{ox}$ and $k_{dd}$. For the calculation of the acid-catalyzed ring-opening reaction rate ($k_{H^+}$), the particle acidity is a 24-h average value estimated using E-AIM II and the equations in Appendix B. As shown in Figure 1, the differences among the factions of IEPOX and MAE to go through the three degradation pathways under the three meteorological conditions are insignificant, especially for MAE.

Hofzumahaus, A., Rohrer, F., Lu, K., Bohn, B., Brauers, T., Chang, C.-C., Fuchs, H., Holland, F., Kita, K., Kondo, Y., Li, X., Lou, S., Shao, M., Zeng, L., Wahner, A., Zhang, Y.: Amplified trace gas removal in the troposphere, Science, 324, 1702–1704, https://doi.org/10.1126/science.1164566, 2009

5. Appendices, Line 476, "Simulation time was set to be 100 h to ensure the completion of reactions." Why the simulation time was set to 100 h? Also, why the completion of reactions was assumed in the simulations? How the variation of the parameters during the day and over the year would affect the simulations?

Response: As we mentioned above, the main objective of using Kintecus kinetic model is to investigate the fate of IEPOX and MAE in the atmosphere and to quantitatively assess the relative importance of the three removal pathways to their decay. As shown in Figure S5 in Eddingsaas et al. (2010), the relative contributions of each pathway to the decay of IEPOX and MAE remained unchanged after these two isoprene intermediates were completely reacted in the atmosphere. Therefore, here we just followed Worton et al. (2013) to set the reaction time of 100 h, which was to make sure that IEPOX and MAE in both gas and aerosol phase had been completely consumed. Actually, the duration time can be other than 100 h, as long as it is long enough for both IEPOX and MAE to be 100% reacted. For example, if it was just for IEPOX, 30 h is pretty enough (Eddingsaas et al., 2010).

Eddingsaas, N.C., Vandervelde, D.G., Wennberg, P.O.: Kinetics and products of the acid-catalyzed ring-opening of atmospherically relevant butyl epoxy alcohols, J. Phys. Chem. A, 114, 8106–8113, https://doi.org/10.1021/jp103907c, 2010.

Worton, D.R., Surratt, J.D., LaFranchi, B.W., Chan, A.W.H., Zhao, Y., Weber, R.J., Park, J.-H., Gilman, J.B., De Gouw, J., Park, C., Schade, G., Beaver, M.R., St. Clair, J., Crounse, J.D., Wennberg, P., Wolfe, G.M., Harrold, S., Thornton, J., Farmer, D., Docherty, K.S., Cubison, M., Jimenez, J.L., Frossard, A., Russell, L.M., Kristensen, K., Glasius, M., Mao, J., Ren, X., Brune, B., Browne, E.C., Pusede, S., Cohen, R.C., Seinfeld, J.H., Goldstein, A.H.: Observational insights into high- and low-NOx aerosol formation from isoprene, Environ. Sci. Technol., 47, 11403–11413, https://doi.org/10.1021/es4011064, 2013.

6. Line 212, "This seasonal trend of monoterpene SOA tracers may be partly due to the lower mixing height and temperature during autumn/winter, which favored the partition of Mono_SOA tracers onto the aerosol phase." Has this correction applied to all samples?

Response: Concentrations of all organic compounds reported in this study, including SOA tracers, were experimentally quantified by GC-MS analysis of $PM_{2.5}$ samples. Therefore, there is no need

to make any corrections to the results. Here we just want to point out that the lower temperature and mixing height can favor the gas-particle partitioning of the semi-volatile Mono_$_{SOA}$ tracers to PM$_{2.5}$ and may be the reason for the observed higher levels of these compounds during winter than summer.

7. Line "353, "It has been well noted that results obtained from this tracer-based method are subject to potential uncertainties from various aspects, e.g., the larger variation of precursor concentrations and more complicated environmental conditions in the real atmosphere than in smog chamber experiments, the decay of some tracer compounds during transport, mismatch of ambient and smog chamber generated SOA compositions, using surrogates other than ketopinic acid for the quantification of tracer compounds, and so on (Ding et al., 2014; Hu et al., 2008; Kleindienst et al., 2012, 2007)." Could the authors comment more quantitively how these factors would affect the results in this study?

Response: It is difficult or impossible to provide a quantitative estimation of how much these above-mentioned factors can affect the TBM results. Because the real environment is much more complicated than the smog-chamber system, and it varies on each sampling day, e.g., the levels of VOC precursors and gaseous oxidants, temperature, RH, solar radiation, and so on. Although the TBM results are subject to potential uncertainties from various aspects, at least it can let us have a rough estimation of the key SOA precursors in the region and their contributions to ambient OC. In this study, SOC$_{TMB}$ showed stark differences under the three meteorological conditions, and it had a similar temporal variation as SOC$_{PMF}$. Moreover, as we mentioned in lines 367-370 in the manuscript, many research groups have adopted this tracer-based method to assess SOC productions from the five studied VOCs at various locations globally, and reasonable results have been obtained.

**Reviewer #3**

**Overview:**

Cheng et al. performed comprehensive chemical analyses on 49 air filters collected in Hong Kong using GC-MS. They used chemical tracers and standards to characterize the organic species from the filters and then applied both PMF and TBM methods to perform source apportionment. Their

results show that the temporal trend of these two methods agree with each other however TBM factors only account for a small portion of the PMF factors. They also show that IEPOX pathway is the dominant pathway to form isoprene-derived SOA during their sampling period by combining chemical characterization with box modeling. Lastly, the team performed correlation analyses to examine the correlation between each SOC factors and common ambient chemicals such as ozone, nitrate, particle sulfate, and acidity and showed different compositions of the SOC may be affected by different chemical species. Overall the manuscript is well written and the chemical analyses are rigorous. The manuscript is recommended for minor revision. I have the following comments for the authors to consider before publication.

Response: Thanks for the reviewer's positive comments.

**Major Comments:**

1. The first major comment I have is mainly related to the description of the PMF method and how they are related to the conclusions. Did the author use the EPA PMF or the Igor based PMF? Which version of the software did the author use? Why the uncertainties of OC/EC and other tracers were set to 20% and 40%, respectively? In addition, any intermediate results on the PMF showing the optimal number of factors to use should be six instead of other values? The author could provide more information about PMF in the SI section to further validate the analysis.

Response: We used the US EPA PMF 5.0, and this information has been updated in the revised manuscript. We followed our previous method to set the uncertainties of the input chemical species in PMF (Hu et al., 2010; Ma et al., 2016, 2019; Cheng et al., 2021). As explained in Hu et al., 2010, there are other sources of uncertainty, such as sampling variability and temporal variation of PM source profiles, in addition to analytical method uncertainties. These sources of errors could not be readily estimated with an equation. Therefore, in the PMF analysis, the uncertainty of each species was adjusted to a uniform value that was equal to the mean analytical uncertainty for this given species in the data set. This, in effect, increased the uncertainties for the low concentration data while the uncertainties for the high concentration data were similar to their corresponding analytical method uncertainties. Such adjustments in uncertainty values have improved the interpretability and therefore adopted in the final PMF analysis.

As we described in section 3.2, for the selection of the optimal number of PMF factors, we first performed preliminary PMF runs with 4 to 8 factors. A hundred base runs were performed in each

modeling run, and the run with the minimum Q value was selected. Besides the minimum Q value, we also used the interpretability and physical meaning of the resolved source profiles as a criterion. Figure 2 below shows the preliminary PMF results with 5 and 7 factors. In the 5-factor solution, hopanes (the marker for vehicle exhaust) was resolved into two factors by about half-half. One is factor 1, with a high loading of levoglucosan, and the other is factor 3 with Ni and V. Therefore, in this 5-factor solution, the vehicle emission factor could not be resolved, and it also messes up the identification of biomass burning and marine vessel factors. In the 7-factor solution, biogenic SOA markers were resolved into two factors (factors 2 and 7), which made the identification of the SOA factor unclear. This confirms that the six-factor solution is the most suitable one.

We have performed a thorough and careful QA/QC check of the final six-factor PMF results. As we presented in the manuscript section 3.2, the interpretability of PMF resolved factors (Figure 2 in the manuscript), the $Q_{Robust}/Q_{True}$ ratio, the distribution of scaled residuals, results of bootstrap runs and DISP assessment, and the strong linear correlation between the measured and PMF-predicted OC ($R^2=0.92$) all indicated that the PMF solution was reliable and robust.

(a)  (b)

[Figure]

Figure 2. PMF results with (a) five factors and (b) seven factors.

2. Similarly, when the author characterized the filter samples into three different categories, i.e., the local sources, the long regional transport, regional source, the back trajectory data and the related information should be provided in the SI as well.

Response: Please refer to our response to Q3 from Reviewer #2.

3. Another comment I have is about the correlation analyses. The author did both Pearson R analysis and the multivariant linear regression (MLR) analysis. The correlation factors in these two analyses do not agree with each other for many categories based on the results in Table 3 and 4. Can the author provide some discussions to explain why the correlation factors from these two analyses do not agree with each other, or even showing opposite trends? And how would the author determine which values to trust?

Response: Sorry that a wrong Table 4 was included in the original manuscript, which does not match the MLR analysis discussion in the manuscript. We are sorry about this mistake, and the correct Table 4 has been updated in the revised manuscript.

We used Pearson's R analysis to evaluate the statistical relationship between two variables and to see how the individual parameters (i.e., $O_3$, $NO_2$, $SO_2$, $O_X$, $NO_3$, sulfate, particle acidity ($H_P^+$), and particle liquid water content ($LWC_P$)) influence each SOC term. The results showed that one SOC term (dependant variable) could be significantly associated with multiple parameters (factors). Therefore, we used multivariate linear regression (MLR) to examine how multiple independent factors contribute to the dependent variable (i.e., the SOC term). It is a common practice in MLR analysis that we should focus on the best possible factors that contribute well to the dependent variable. Results from Pearson's R analysis showed that $SO_2$ was not significantly associated with $SOC_{PMF}$, $SOC_{SOA}$, and $SOC_{TBM}$. Therefore, $SO_2$ was excluded from the MLR analysis. Since $O_X$ = $NO_2$ + $O_3$, it is not an independent factor and should not be included in the MLR analysis together with $NO_2$ and $O_3$. That is why $O_X$ was not included in MLR either. As we mentioned in lines 439 to 442 in the revised manuscript, "Six parameters, namely $O_3$, $NO_2$, $NO_3$, sulfate, $H_P^+$, and $LWC_P$, were included in the preliminary runs. However, the MLR results showed that $O_3$ was an insignificant factor for all SOC terms, even with negative regression coefficients. Pearson's R analysis also showed that SOC was more $NO_2$ dependent than $O_3$. Therefore, it was excluded from the final MLR analysis, and the results were shown in Table 4."

It is not surprising that MLR and Pearson's R analysis results do not 100% agree with each other. As we just discussed, Pearson's R analysis evaluates the correlation between a factor and a SOC term, and one R-value is obtained in each analysis. While MLR examines how multiple independent factors influence the SOC term and multiple β-coefficients are determined in one analysis. MLR gives us a rough estimation of how significant can the factors affect SOC formation and to what extent.

4. My last major comment is about the sample collection time. The filter sample collection spans almost a year. The VOC emissions and atmospheric oxidation chemistry might be quite different for summer and winter seasons. Would combine all the samples from different seasons together miss any seasonal variabilities and lead to misinterpretation of the data?

Response: No samples are combined. A total of 49 samples were analyzed. When we did the PMF, Pearson's R, and MLR analysis, experimental data from each $PM_{2.5}$ sample were used. Moreover, Hong Kong is a typical coastal megacity located in the Pearl River Delta region with an annual average temperature of 24.17±5.00 $^o$C. Our previous studies on organic aerosols and HULIS in Hong Kong (Hu et al., 2008, 2010, 2013; Ma et al., 2019, 2020) all showed that variations of the levels and sources of ambient organic aerosols in Hong Kong are more influenced by air mass origins than temperature and solar radiation.

Hu, D., Yu, J. Z.: Secondary organic aerosol tracers and malic acid in Hong Kong: Seasonal trends and origins, Environ. Chem., 10, 381–394, https://doi.org/10.1071/EN13104, 2013.

Hu, D., Bian, Q., Li, T.W.Y., Lau, A.K.H., Yu, J.Z.: Contributions of isoprene, monoterpenes, β-caryophyllene, and toluene to secondary organic aerosols in Hong Kong during the summer of 2006, J. Geophys. Res. Atmos., 113, https://doi.org/10.1029/2008JD010437, 2008.

Hu, D., Bian, Q., Lau, A.K.H., Yu, J.Z.: Source apportioning of primary and secondary organic carbon in summer $PM_{2.5}$ in Hong Kong using positive matrix factorization of secondary and primary organic tracer data, J. Geophys. Res. Atmos., 115, 1–14, https://doi.org/10.1029/2009JD012498, 2010.

Ma, Y., Cheng, Y., Qiu, X., Cao, G., Kuang, B., Yu, J.Z., Hu, D.: Optical properties, source apportionment and redox activity of humic-like substances (HULIS) in airborne fine particulates in Hong Kong, Environ. Pollut., 255, 113087, https://doi.org/10.1016/j.envpol.2019.113087, 2019.

Ma, Y., Cheng, Y., Cao, G., Yu, J.Z., Hu, D.: Speciation of carboxylic components in humic-like substances (HULIS) and source apportionment of HULIS in ambient fine aerosols ($PM_{2.5}$) collected in Hong Kong, Environ. Sci. Pollut. Res., 27, 23172-23180, doi:10.1007/s11356-020-08915-w, 2020.

**Minor Comment:**

5. L19: Should SOC be defined first?

Response: Revised.

6. L80: the sample collection

Response: We can not find this term in L80.

7. L223: "b" in beta should be lower case and "C" in Caryophyllinic should be upper case. Beta should be italic as well.

Response: Revised.

8. The categories in Table 3 and 4 should be in the same order for easy comparison.

Response: Table 4 has been updated in the revised manuscript.